# AIRE is a critical spindle-associated protein in embryonic stem cells

Bin Gu[1], Jean-Philippe Lambert[2], Katie Cockburn[1], Anne-Claude Gingras[2,3], Janet Rossant[1,3]*

[1]Program in Developmental and Stem Cell Biology, Hospital for Sick Children, Toronto, Canada; [2]Lunenfeld-Tanenbaum Research Institute at Mount Sinai Hospital, Toronto, Canada; [3]Department of Molecular Genetics, University of Toronto, Toronto, Canada

**Abstract** Embryonic stem (ES) cells go though embryo-like cell cycles regulated by specialized molecular mechanisms. However, it is not known whether there are ES cell-specific mechanisms regulating mitotic fidelity. Here we showed that Autoimmune Regulator (*Aire*), a transcription coordinator involved in immune tolerance processes, is a critical spindle-associated protein in mouse ES(mES) cells. BioID analysis showed that AIRE associates with spindle-associated proteins in mES cells. Loss of function analysis revealed that *Aire* was important for centrosome number regulation and spindle pole integrity specifically in mES cells. We also identified the c-terminal LESLL motif as a critical motif for AIRE's mitotic function. Combined maternal and zygotic knockout further revealed Aire's critical functions for spindle assembly in preimplantation embryos. These results uncovered a previously unappreciated function for *Aire* and provide new insights into the biology of stem cell proliferation and potential new angles to understand fertility defects in humans carrying *Aire* mutations.

*For correspondence: janet. rossant@sickkids.ca

**Competing interests:** The authors declare that no competing interests exist.

## Introduction

Self-renewal capability, defined as the ability of cells to proliferate while sustaining differentiation potential, is one of the defining features of stem cells (*Martello and Smith, 2014*). Robust proliferation ability, not only rapid cell division but also stable karyotype maintenance, ensures the maintenance and expansion of stem cell populations. Embryonic stem (ES) cells, the in vitro counterpart of preimplantation epiblast cells (*Boroviak et al., 2014*; *Boroviak and Nichols, 2014*), possess particularly robust self-renewal capability (*Martello and Smith, 2014*). Unlike most somatic stem cell types that divide relatively infrequently, ES cells undergo constant proliferation while accumulating few karyotypic abnormalities (*Suda et al., 1987*; *Weissbein et al., 2014*). ES cells undergo specialized embryo-like cell cycles, characterized as rapid cell cycles with brief gap phases (*Kareta et al., 2015*; *Coronado et al., 2013*). Unlike in most somatic cells where coordinated fluctuation of Cyclin-CDK activities drive the cells through the cell cycle, ES cells possess higher and constant activity of most Cyclin-CDK pairs except for Cyclin B-CDK1 (*Stead et al., 2002*). These cell cycle patterns have been proposed to limit the window in which ES cells are responsive to differentiation cues, therefore promoting self-renewal (*Dalton, 2015*; *Pauklin and Vallier, 2013*). However, it also subjects the mitosis process to additional stress since the fluctuating activity of Cyclin-CDKs is vital for aspects of mitotic fidelity, such as centrosome maturation/amplification and kinetochore microtubule stabilization (*Lacey et al., 1999*; *Haase et al., 2001*; *Chen et al., 2002*). Given that ES cells actually show remarkable karyotypic stability, this might suggest that they possess additional specific molecular mechanisms to ensure mitosis fidelity.

**eLife digest** Before the cells in our body separate to create copies of themselves, they need to duplicate their genetic material. To do so, they construct a machine called the spindle apparatus to divide their DNA evenly. In the embryo of mammals, embryonic stem cells – cells that can create all the cell types in adults – divide swiftly. This makes them prone to make mistakes that could affect the health of the organism they will develop into. To limit the number of errors, the embryonic stem cells divide more stringently and have more effective machineries to create a spindle apparatus.

Together with a team of researchers, Gu previously showed that a gene and its protein called the 'autoimmune regulator', or AIRE for short, are highly active in embryonic stem cells and embryos. The autoimmune regulator usually plays an important role in helping immune cells to distinguish the body's own proteins from those of foreign invaders. It was also shown that some people suffering from fertility problems carry mutations in the AIRE gene. However, until now, it was not known if AIRE also had a specific role in embryonic stem cells.

Using mouse embryonic stem cells and early embryos that were modified to either completely lack the AIRE gene or to produce a defective AIRE protein, Gu et al. now discovered a new purpose for AIRE. The AIRE protein interacted with a group of proteins that are associated with the spindle apparatus and was needed so that the spindle could form properly. When cells lacked the AIRE gene, a faulty spindle apparatus was assembled. Moreover, when a defective AIRE protein was produced, the structure of spindle apparatus in embryonic stem cells was also disrupted.

A next step will be to further investigate how AIRE mutations could affect stem cell maintenance and fertility, which could lead to better ways to detect fertility defects in the future.

Autoimmune Regulator (*Aire*) is a central coordinator of immune tolerance. It is specifically expressed in medullary thymic epithelium cells (mTECs) and mediates negative selection against auto-reactive T cells by inducing promiscuous expression of tissue specific antigens (*Anderson et al., 2002*; *Ramsey et al., 2002*). However, it has long been known that *Aire* is also expressed in germ cell progenitors and that mutations of *Aire* could cause fertility defects in mouse and human (*Finnish-German APECED Consortium et al., 1997*; *Hubert et al., 2009*; *Schaller et al., 2008*). Fertility defects were frequently attributed to aneuploidy in embryo cells, originating from meiotic or mitotic errors (*Tempest and Martin, 2009*; *Martin, 2008*; *Webster and Schuh, 2017*; *Daughtry and Chavez, 2016*), implying a possible role of Aire in these processes. We and others have reported the expression of *Aire* in mouse ES cells and early embryos (*Gu et al., 2010*; *Nishikawa et al., 2010*; *Bin et al., 2012*). The expression of *Aire* was specific to undifferentiated mouse ES cells and gradually diminished with differentiation at both mRNA and protein levels (*Gu et al., 2010*). Among embryo-derived stem cell lines, *Aire* expression was specific to pluripotent ES cells and not found in extraembryonic trophoblast stem cells (the counterpart of postimplantation extraembryonic ectoderm(EXE) progenitor cells) (*Roberts and Fisher, 2011*) or eXtraembryonic Endoderm cells (our unpublished data) at both mRNA and protein levels. Interestingly, *Aire* has also been shown to be up-regulated during the final stage of induced pluripotent stem cell (iPS) formation when iPS clones become transgene-independent (*Hussein et al., 2014*). The presence of *Aire* mRNA has been detected in mouse oocytes and all preimplantation stages and early postimplantation stages(up till E6.5) embryos but the expression levels and patterns at the protein level are unknown (*Nishikawa et al., 2010*). Moreover, it was recently shown that *Aire* mRNA is highly expressed in peri-implantation (E4.5–5.5) mouse epiblast cells (*Boroviak et al., 2014*; *Chen et al., 2016*), the in vivo cell type that naïve ES cells most likely represent. We have previously shown that knocking down *Aire* in mouse ES cells caused self-renewal and karyotype defects, suggesting a role for *Aire* in mitosis (*Gu et al., 2010*). However, the underlying mechanism remained elusive.

AIRE is not a canonical transcription factor that recognizes specific DNA sequence motifs (*Mathis and Benoist, 2009*), and so there is considerable interest in understanding how AIRE functions in inducing promiscuous expression and other processes. Identifying its interaction partners is critical for revealing the molecular pathways in which AIRE is involved. Through different methods including GST pulldown, yeast two-hybrid and co-immunoprecipitation followed by mass

spectrometry, a number of interacting partners, including DAXX, P-TEFb and ATF7ip-MBD1, have been identified for AIRE (*Abramson et al., 2010*; *Meloni et al., 2010*; *Oven et al., 2007*; *Waterfield et al., 2014*). The general conclusion from these studies is that AIRE performs its transcriptional function by acting as a hub protein, coordinating chromatin remodeling, general transcription, RNA processing and nuclear transport, to induce promiscuous gene expression (*Abramson et al., 2010*). However, most of these studies were performed in somatic cell lines lacking endogenous *Aire* expression, which may not reveal its natural interactome. Moreover, the co-immunoprecipitation methods used in most of these studies are more efficient in identifying soluble proteins, rather than proteins tightly bound to insoluble structures like condensed chromosomes or mitotic spindles (*Lambert et al., 2015*). Other approaches are more suited for capturing partners of insoluble components. BioID is an in vivo proximity biotinylation based assay for identifying proximity partners of a protein of interest. In this assay the protein of interest (bait protein) is expressed as a fusion protein with an abortive biotin ligase (BirA R118G or BirA*) which effectively catalyzes the formation of locally-concentrated activated biotin that covalently labels proximal proteins, the labeled proteins can be recovered under harsh solubilizing conditions, purified by avidin affinity purification and then identified by mass spectrometry analysis. BioID has been employed to identify proximity interactions in relatively insoluble structures including chromatin, nuclear envelope and centrosomes (*Roux et al., 2012*; *Lambert et al., 2015*; *Gupta et al., 2015*; *Kim et al., 2014*). We reasoned that identification of AIRE's partners with BioID methods in ES cells could help reveal its full function in stem cells and uncover new regulatory aspects of stem cell self-renewal.

We characterized the proximal protein partners of AIRE in mouse ES(mES) cells using BioID technology and found that, besides proteins functioning in known AIRE-related processes such as general transcription and RNA processing, AIRE also interacts with a group of mitotic spindle-associated proteins. We present evidence that AIRE is a spindle-associated protein in mES cells where it is essential for spindle assembly, centrosome number and structural maintenance. We further showed that the last LxxLL(LESLL) motif, a multifunctional motif that has been implicated in mediating protein-protein interactions (*Plevin et al., 2005*), is critical for the mitotic function of AIRE in mES cells and that a known human disease mutation specifically affecting this motif could induce mitotic defects. We also present data showing that maternal-zygotic *Aire* knockout embryos have mitosis defects at the blastocyst stage. These results provide a new insight into *Aire*'s non-immune functions in stem cells and suggest novel ES-specific mechanisms for regulating mitotic fidelity.

## Results

### AIRE is a spindle-associated protein in mES cells

To identify the interaction partners of AIRE in ES cells, we conducted BioID analysis (*Figure 1A*) (*Roux et al., 2012*). ES cells expressing *mCherry-BirA*[*] were used as controls to exclude promiscuous biotinylation activities of the BirA[*] enzyme and *Aire-BirA** expressing ES cell samples were used to identify interaction partners. As shown by streptavidin-FITC staining in *Figure 1B*, mCherry-BIRA[*] biotinylated proteins distributed evenly in cells, while AIRE- BIRA[*] biotinylated proteins showed an interesting mitotic spindle association (*Figure 1B* arrowheads and magnified images). Western blotting analysis confirmed that the expression level of mCherry-BIRA[*]/AIRE-BIRA[*] and biotinylated protein amounts were similar between controls and samples (*Figure 1—figure supplement 1A*). The biotinylated proteins were then subjected to LC-MS/MS analysis and a high confidence list of AIRE-interacting partners was generated using the SAINT algorithm with FDR $\leq$ 1% (*Figure 1C*, *Figure 1—source data 1*) (*Choi et al., 2011*; *Teo et al., 2014*). Strikingly, Gene Ontology (GO) analysis of AIRE-interacting partners revealed that the two most enriched Biological Processes (BP) were 'mitotic nuclear division' and 'cell division' (*Figure 1—figure supplement 2*). As for Cellular Compartment (CC), mitosis-related terms such as 'cytoskeleton', 'microtubule' 'centrosome', 'spindle pole' and 'spindle' were enriched(*Figure 1—figure supplement 2*). The interactors included proteins functioning in three crucial aspects of mitosis (*Figure 1C*): (i) kinase cascade regulation of the mitotic process (eg. AURKB, CDK2) (*Wieser and Pines, 2015*), (ii) centrosome/spindle pole maturation and integrity (eg. SPICE1, HAUS5, HAUS8) (*Fu et al., 2015*; *Comartin et al., 2013*; *Lawo et al., 2009*) and (iii) regulation of microtubule dynamics in spindles (eg. CKAP2, CLASP1, CLASP2) (*Bratman and Chang, 2008*). These data suggested spindle localization and mitosis-

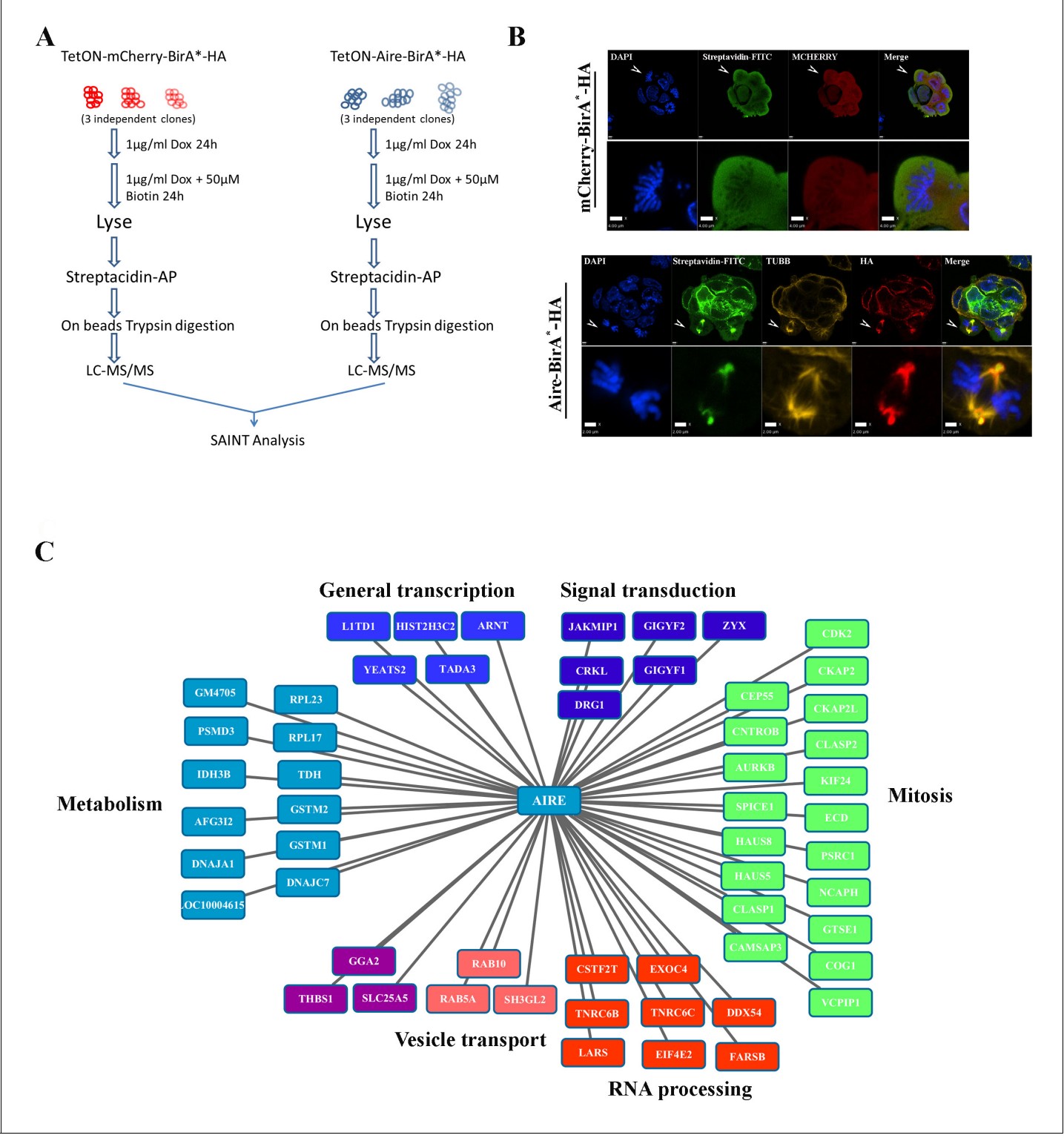

**Figure 1.** Proximity partners of AIRE in ES cells. (**A**) Flow chart of BioID experiments. mCherry-BirA*-HA expressing ES cells were used as control to subtract the effect of promiscuous biotinylation activity of BirA*. (**B**) Localization of bait proteins (mCherry or HA) and biotin-tagged proteins (Streptavidin-FITC) in mCherry-BirA*-HA control ES cells and Aire-BirA*-HA ES cells shown by immunofluorescence staining. Scale bars: 1 μm. Bottom panel for each group shows magnified views of a mitotic cell, marked by arrows in top panel. Scale bars: 4 μm for mCherry-BirA*-HA, 2 μm for Aire-BirA*-HA. (**C**) List of proximity partners of AIRE identified by SAINT analysis (n = 3, 1%FDR) of BioID/mass spectrometry data and annotations of functional categories.

*Figure 1 continued on next page*

*Figure 1 continued*

The following source data and figure supplements are available for figure 1:

**Source data 1.** List of proximity partners of AIRE.

**Figure supplement 1.** Validations of BioID analysis.

**Figure supplement 2.** Gene ontology enrichment analysis of AIRE's proximity partners with DAVID.

associated functions of AIRE in ES cells. Additionally, proteins functioning in general transcription processes and RNA processing were also identified, suggesting that *Aire* may also play a similar role in ES cells as in immune cells (*Figure 1C*). Considering the proximity ligation nature of BioID (~10 nm [*Kim et al., 2014*]), we performed Duolink Proximity Ligation Assay (PLA) (*Söderberg et al., 2006*) and validated a number of interaction partners of AIRE in ES cells. Duolink signal between anti-Flag M2 antibody and antibodies against respective candidate partners in mES cells before doxycycline induction of 3XFlag-AIRE were used to control background signal and Duolink signal between 3XFlag-AIRE and AURKA, a protein not identified as AIRE's proximity partner but highly related to an identified partner AURKB, was used as negative control. As shown in *Figure 1—figure supplement 1B*, six AIRE proximity partners showed strong Duolink signal over background while AURKA only displayed background level signals,

The identification of mitotic interaction partners of AIRE in mES cells prompted us to examine the subcellular localization of AIRE in mES cells undergoing mitosis. Immunostaining with an antibody against mouse AIRE showed co-localization of AIRE with α-tubulin (TUBA)-marked mitotic spindles (*Figure 2A*). To further corroborate this finding, we generated *Aire*-3XFlag mice (C-terminal 3XFlag-Tag fusion to endogenous *Aire*) using CRISPR-Cas9 technology and derived ES cells from these mice. Co-staining with the M2 anti-Flag antibody and β-tubulin(TUBB) antibody showed that AIRE co-localized with the mitotic spindle apparatus (*Figure 2—figure supplement 1*). To gain further insight into the precise localization of AIRE on the spindle, we performed Structured Illumination Microscope (SIM) analysis. We found AIRE presented as localized foci along the spindle microtubules during metaphase and anaphase (*Figure 2B*). It also formed a cloud-like structure covering the spindle pole and its peripheral region, suggesting a role in spindle pole/centrosome functions. In mitotic telophase cells, AIRE localized to the base of the mid-spindle, sparing the central part of midbody (*Figure 2B*), indicating a possible role in cytokinesis. These results identify AIRE as a bona fide spindle associated protein during mitosis in ES cells, in addition to its earlier reported nuclear foci localization in interphase cells (*Gu et al., 2010*).

We then performed domain mapping to identify the domains required for the spindle localization of AIRE. Wild type and six truncated forms of Flag-tagged AIRE (*Figure 2C*) were introduced into ES cells as doxycycline (Dox) inducible transgenes and the spindle localization was investigated following addition of Dox by immunofluorescence staining. We found that the removal of either the HSR/CARD domain or the SAND domain abrogated the spindle localization of AIRE, while deletions of any of the two PHD domains or the C-terminal 70 amino acid(aa) flexible tail region (AIRE$^{\Delta c70}$) had no effect on localization to the spindle (*Figure 2C*). However, defective spindle morphology and decreased colony formation ability were observed upon 12 hr overexpression of the AIRE$^{\Delta c70}$ truncated form, indicating a dominant negative effect on spindle assembly (*Figure 2—figure supplement 2A*). The expression of core pluripotency factors OCT4, NANOG and SOX2 was not changed at this time point (*Figure 2—figure supplement 2B*). Therefore the two N-terminal domains HSR/CARD and SAND were critical for the spindle recruitment of AIRE while the last 70aa were essential for its spindle-associated function.

## Aire is essential for spindle assembly in ES cells

We were not able to generate homozygous *Aire* null ES clones directly through CRISPR-Cas9 mediated knockout, likely due to an essential role of Aire in ES cells. In order to conduct Aire loss of function analysis, we generated an *Aire* floxed (*Fl*) mouse line using CRISPR-Cas9 technology (*Figure 3—*

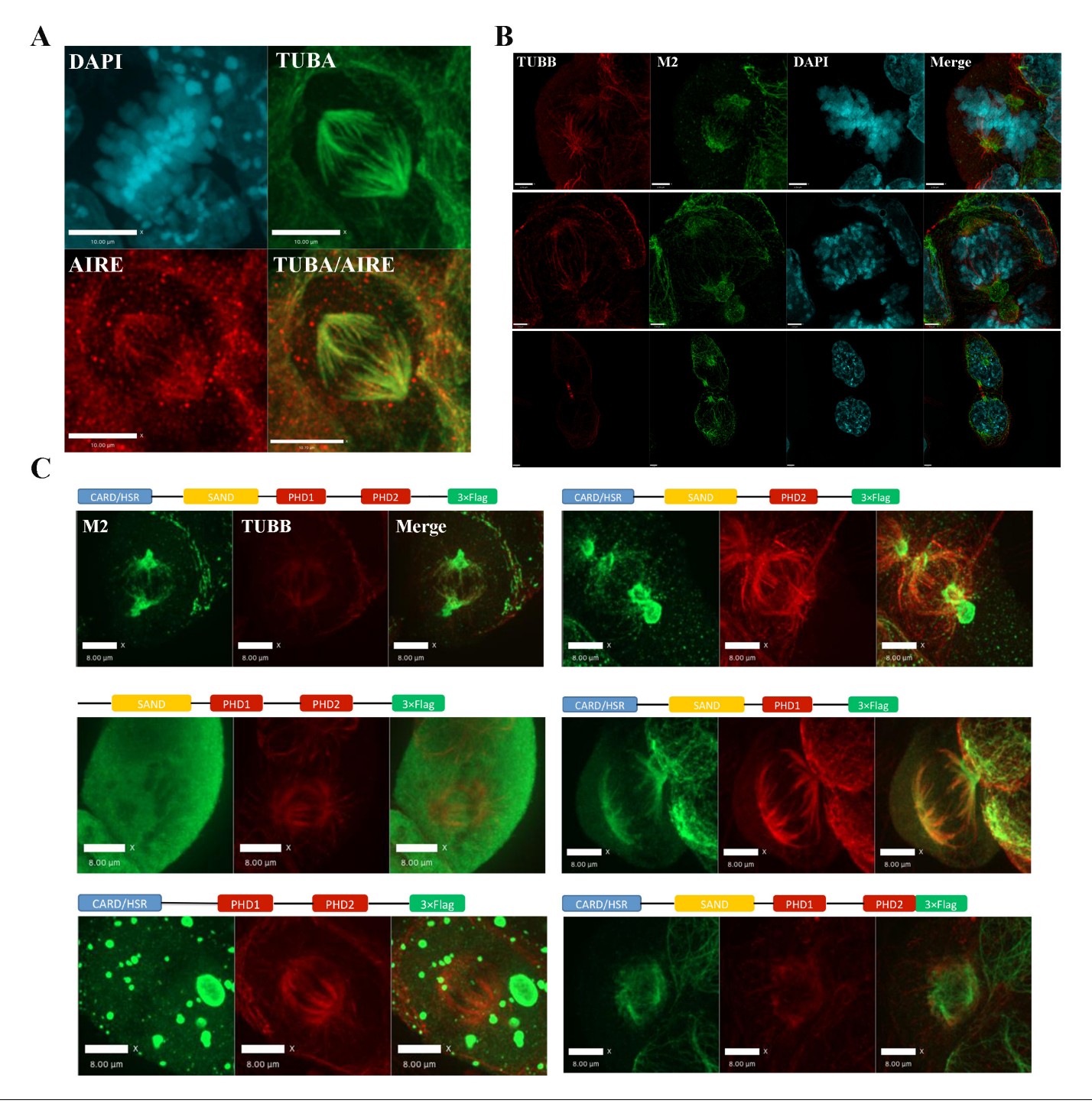

**Figure 2.** AIRE localizes to mitotic spindles in ES cells. (**A**) Immunofluorescence staining of AIRE and α-tubulin (TUBA) in AmES8 cells. Scale bar: 10 μm. (**B**) Structural Illumination Microscope imaging of spindle localization of AIRE in AIRE-3XFlag expressing ES cells during different phases of mitosis. The Flag-tag was detected using the M2 mouse monoclonal antibody and spindle microtubules were marked by β-tubulin (TUBB). Scale bar: 2.5 μm. (**C**) Localization of different truncated versions of Aire-3XFlag (shown above each image) in mitotic ES cells by immunofluorescence staining. Scale bar: 5 μm.

The following figure supplements are available for figure 2:

**Figure supplement 1.** 3XFlag-AIRE localized to mitotic spindles in Aire-3XFlag mES cell.

*Figure 2 continued on next page*

Figure 2 continued

**Figure supplement 2.** AIRE$^{\Delta c70}$ functions as a dominant negative, disrupting spindle assembly and inhibiting proliferation in mES cells.

figure supplement 1). The mice were then bred with *R26CreERT* mice and ES cell lines were derived. A pair of *Aire$^{+/+}$;R26CreERT$^{+/+}$* and *Aire$^{Fl/Fl}$;R26CreERT$^{+/+}$* (Pair 1) (*Figure 3*) and a pair of *Aire$^{Fl/+}$;R26CreERT$^{±}$* and *Aire$^{Fl/Fl}$;R26CreERT$^{+/-}$* (Pair 2) (*Figure 3—figure supplement 1*) ES cell clones were used for the experiments. 4OH-Tamoxifen(TAM) treatment induced recombination only in *Aire$^{Fl/Fl}$;R26CreERT$^{+/+}$* cells (from hereon referred to as *Aire$^{-/-}$* cells) (*Figure 3A*). The knockout of the targeted region (exons 8–9) was validated by Sanger sequencing. All control lines (*Aire$^{+/+}$; R26CreERT$^{+/+}$*, *Aire$^{+/+}$;R26CreERT$^{+/+}$* + TAM and *Aire$^{FL/FL}$;R26CreERT$^{+/+}$*) were normal and showed no changes in gene expression, morphology or colony formation. For simplicity we refer to all control lines as *Aire$^{+/+}$* cells in the main text from here on. qPCR analysis showed the loss of the expression of *Aire* mRNA containing Exon 8 and 9 in *Aire$^{-/-}$* cells (*Figure 3B*). A pair of primers against the 3' end of *Aire* mRNA detected similar level of expression in both *Aire$^{+/+}$* and *Aire$^{-/-}$* cells (*Figure 3B*), indicating production of truncated mRNAs. However, the complete loss of AIRE protein expression was validated by immunostaining (*Figure 3C*). Consistent with previous data from Aire knockdown ES cells (*Gu et al., 2010*), the mRNA level of the pluripotency markers *Pou5f1(Oct4)* and *Nanog* were decreased in *Aire$^{-/-}$* cells while no significant difference was detected for *Sox2*. Correspondingly we also detected a decrease of Oct4 and Nanog protein levels in *Aire$^{-/-}$* cells by immuno-fluorescence analysis (*Figure 3—figure supplement 3*). Both proliferation and colony formation capability of *Aire$^{-/-}$* cells were impaired (*Figure 3D*), suggesting a role of Aire in ES cell self-renewal. We investigated the spindle structures by immunofluorescence staining of α-tubulin(TUBA) and γ-tubulin(TUBG) and found defective spindles with multiple γ-tubulin positive foci in *Aire$^{-/-}$* cells (*Figure 3E*). Quantitative analysis showed a significant increase in the number of γ-tubulin foci in *Aire$^{-/-}$* mitotic cells (*Figure 3F*). We obtained similar phenotypes using the pair two conditional *Aire* knockout ES cell lines (*Figure 3—figure supplement 2*).

## LESLL motif is essential for the mitotic function of Aire in ES cells

The LxxLL motif has been shown to mediate protein-protein interactions in many different contexts. There are four LxxLL motifs in AIRE protein (Uniprot Q9Z0E3), with the last one (LESLL) within the last 70aa of the protein (*Figure 4A*). Since the c-terminal 70aa truncated form of AIRE produced a dominant negative effect on spindle assembly, we examined whether this motif was essential for the mitotic function of AIRE. Upon overexpression of full length AIRE protein lacking the LESLL motif (AIRE $^{\Delta LESLL}$), most ES cells rounded up and underwent cell death within 48 hr (*Figure 4B*). Similarly, overexpression of AIRE $^{\Delta LESLL}$ almost completely abolished the colony-forming capability of ES cells (*Figure 4B*). Immunofluorescence analysis demonstrated malformed spindle apparatus in these cells (*Figure 4C*). Flow cytometry analysis using the mitotic marker pS10-H3 revealed a severe mitotic arrest in AIRE $^{\Delta LESLL}$ overexpressing cells (*Figure 4D*). Notably, although also localized to the spindle in mitotic NIH3T3 cells that do not express Aire endogenously, AIRE $^{\Delta LESLL}$ did not induce spindle disruption or mitotic arrest in this cell type (*Figure 4C,D* and *Figure 4—figure supplement 1*).

The spindle assembly checkpoint (SAC) is the main mechanism that arrests cells with defective spindles and subsequently induces apoptosis. We examined the activation state of SAC in AIRE $^{\Delta LESLL}$ overexpressing cells and found that the deposition of BubR1 on mitotic chromosomes was significantly increased (*Figure 4—figure supplement 2A*). Inhibition of the major SAC kinase Mps1 rescued the mitotic arrest induced by AIRE $^{\Delta LESLL}$ overexpression, further indicating that functional activation of SAC is the mechanism behind mitotic arrest and cell death in AIRE $^{\Delta LESLL}$ overexpressing cells (*Figure 4—figure supplement 2B*).

In humans, mutations in *AIRE* cause Autoimmune Polyendocrinopathy-Candidiasis-Ectodermal Dystrophy (APECED), which been associated with infertility in some patients through an elusive mechanism. We searched through known human AIRE mutations to see if any would result in production of AIRE truncated proteins mimicking AIRE $^{\Delta LESLL}$. In 2000, a rare mutation, AIRE 33031delG, was reported from Japan (*Ishii et al., 2000*). The mutation resulted in a frame-shift in the Ala505 codon, an amino acid sequence change thereafter and premature termination at amino

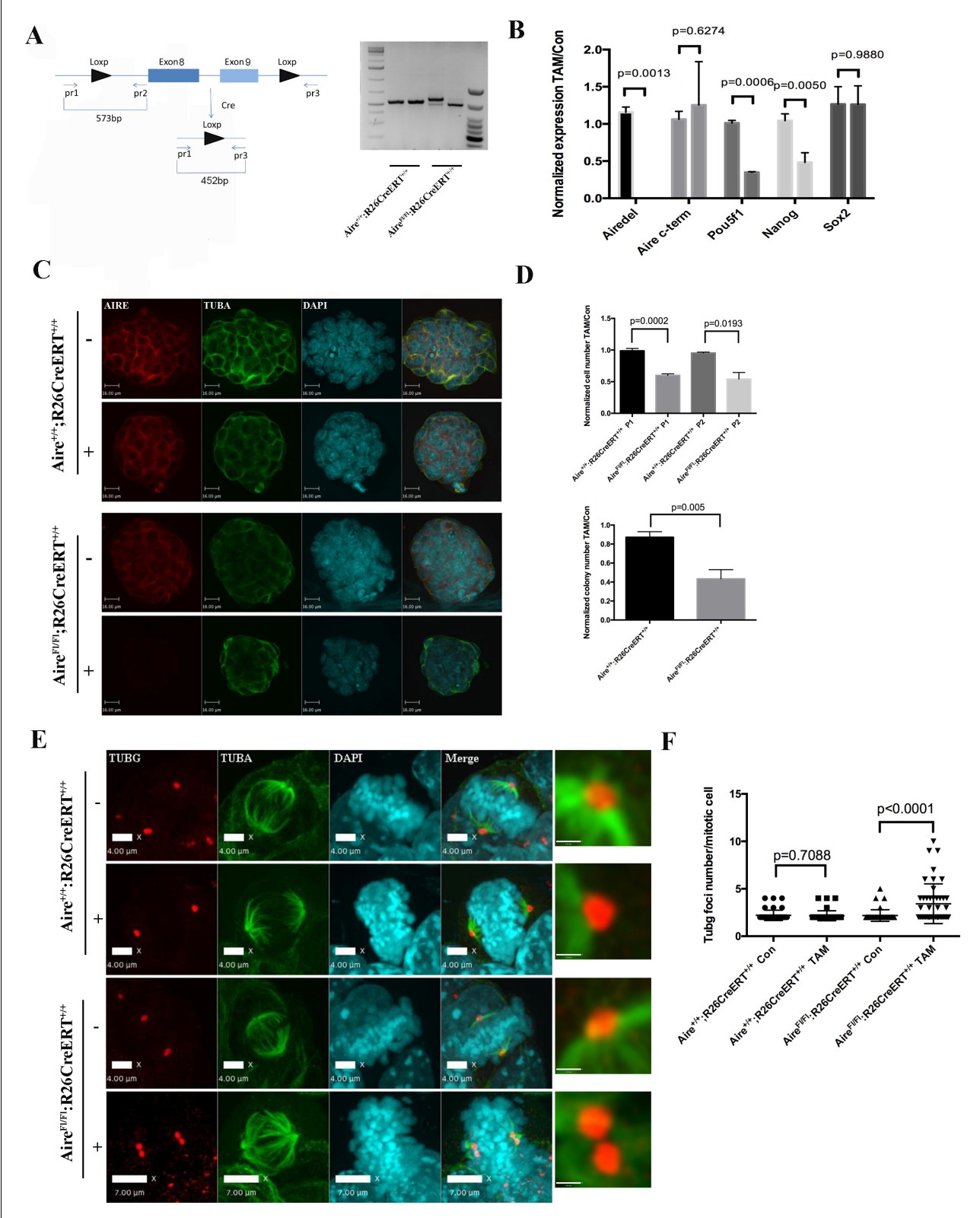

**Figure 3.** Aire is essential for spindle assembly in ES cells. (**A**) Scheme of the *Aire^Fl* allele and genotyping strategy before and after cre-induced deletion. Genotyping results from *Aire^{+/+}R26CreERT^{+/+}* and *Aire^{Fl/Fl} R26CreERT^{+/+}* ES cells without tamoxifen (-) and with 5 ng/ml tamoxifen for 24 hr (+). (**B**) qPCR analysis of gene expression at passage one after tamoxifen treatment in *Aire^{+/+}R26CreERT^{+/+}* and *Aire^{Fl/Fl} R26CreERT^{+/+}* ES cells. Aire del: primers complementary to deleted region of *Aire*. Aire c-term: primers complementary to c-terminal region of *Aire*. Data presented as mean ± sd
*Figure 3 continued on next page*

*Figure 3 continued*

of 3 biological replicates. *p*-values were calculated with Wilson's t-test. (**C**) Validation of *Aire* knockout by immunofluorescence imaging in *Aire*$^{+/+}$*R26CreERT*$^{+/+}$ and *Aire*$^{Fl/Fl}$ *R26CreERT*$^{+/+}$ ES cells without tamoxifen (-) and with 5 ng/ml tamoxifen for 24 hr (+). bar: 16 μm. (**D**) Proliferation over two passages and colony formation assay showing total cell number (top panel) and ALP positive colony number (bottom panel) in tamoxifen treated over non-treated *Aire*$^{+/+}$*R26CreERT*$^{+/+}$ and *Aire*$^{Fl/Fl}$ *R26CreERT*$^{+/+}$ ES cells. Data in each panel presented as mean ± sd of 3 biological replicates. *p*-values were calculated with Wilson's t-test. (**E**) Representative immunofluorescence images of γ-tubulin (TUBG) foci on mitotic spindles in passage 1 *Aire*$^{+/+}$*R26CreERT*$^{+/+}$ and *Aire*$^{Fl/Fl}$ *R26CreERT*$^{+/+}$ ES cells without tamoxifen (-) and with 5 ng/ml tamoxifen for 24 hr (+). Scale bar: 4 μm. Right column shows magnified views of spindle poles. Scale bar: 1 μm. (**F**) Quantitation of γ-tubulin (TUBG) foci on mitotic spindles in passage 1 *Aire*$^{+/+}$*R26CreERT*$^{+/+}$ and *Aire*$^{Fl/Fl}$ *R26CreERT*$^{+/+}$ ES cells without tamoxifen (Con) and with 5 ng/ml tamoxifen for 24 hr (TAM). γ-tubulin (TUBG) foci were counted from 30 mitotic cells from three biological replicates for each group. Each point represents one mitotic cell. Error bars: s.d. of mean. *p*-values were calculated with Mann–Whitney–Wilcoxon test.

The following figure supplements are available for figure 3:

**Figure supplement 1.** Complete diagram of *Aire* floxed (Fl) allele and cre induced depletion of critical exons.

**Figure supplement 2.** *Aire* knockout abrogated proliferation and spindle assembly in Pair 2 ES cells.

**Figure supplement 3.** Immunofluorescence images of pluripotency markers in Aire Floxed;R26CreERT cells.

acid 520, which eliminated the LESLL motif (aa 516–520 (*Figure 4E*). Since the premature termination codon was located 7 bp upstream of the last Exon-Exon junction, the mRNA would likely escape nonsense-mediated decay, resulting in a truncated protein lacking the last LESLL motif. We investigated the function of this mutation in mouse ES cells by overexpressing a murine *Aire* version mimicking the AIRE 33031delG-induced truncation) (AIRE505fs). We found that the truncated protein caused similar spindle defects and mitotic arrest in mES cells (*Figure 4F and G*). Therefore the last LESLL motif is essential for the mitotic function of Aire in mouse ES cells and a human mutant AIRE lacking the c-terminal residues containing LESLL motif could potentially lead to mitotic defects during human reproduction and development.

## Aire is essential for centrosome number maintenance and spindle pole integrity in ES cells

The BioID data showed that Aire established proximal interactions with a number of proteins functioning in centrosome duplication and maturation, including SPICE1, HAUS5 and HAUS8. We also observed extra γ-tubulin positive foci of various sizes on defective spindles in both *Aire*$^{-/-}$ and AIRE$^{ΔLESLL}$ overexpressing cells (*Figures 3* and *4*). Therefore we investigated the centrosome and spindle pole structure under SIM with γ-tubulin/Nedd1 double staining. Interestingly, both *Aire*$^{-/-}$ and AIRE$^{ΔLESLL}$ overexpressing ES cells possessed extra centrosomes in S/G2 phases, while the size of the centrosome was not obviously different (images in *Figure 5A and C* and quantifications in *Figure 5B and D*, *Figure 5—figure supplement 1*). During mitosis, spindle poles with increased size consisting of multiple aggregated centrosomes were frequently observed in *Aire*$^{-/-}$ cells, while in AIRE$^{ΔLESLL}$ overexpressing ES cells, spindle poles were obviously fragmented (images in *Figure 5A and C* and quantifications in *Figure 5B and D*, *Figure 5—figure supplement 1*). Since AIRE$^{ΔLESLL}$ overexpression caused stronger proliferation and mitosis phenotypes, we further investigated the microtubule organization center (MTOC) functions of mitotic centrosomes in these cells. We found that instead of forming bipolar spindles, the spindle microtubules formed a disorganized meshwork with fragmented γ-tubulin rings upon AIRE$^{ΔLESLL}$ overexpression, suggesting the abrogation of MTOC functions (*Figure 5E*). The spindle pole/centrosome defect observed here could be either due to malformed centrioles or to failed organization and maintenance of pericentriolar material (PCM) integrity. Using SIM, we imaged centriole and pericentriolar material by double staining mES cells for acetyl-tubulin and γ-tubulin. We did not observe any obvious morphological defects of centrioles in AIRE$^{ΔLESLL}$ overexpressing cells in both S/G2 and M phases (*Figure 5—figure supplement 2*). However, although the γ-tubulin-marked pericentriolar material compactly surrounded the mitotic centriole in all the control groups, it failed to cover the centriole in mitotic AIRE$^{ΔLESLL}$

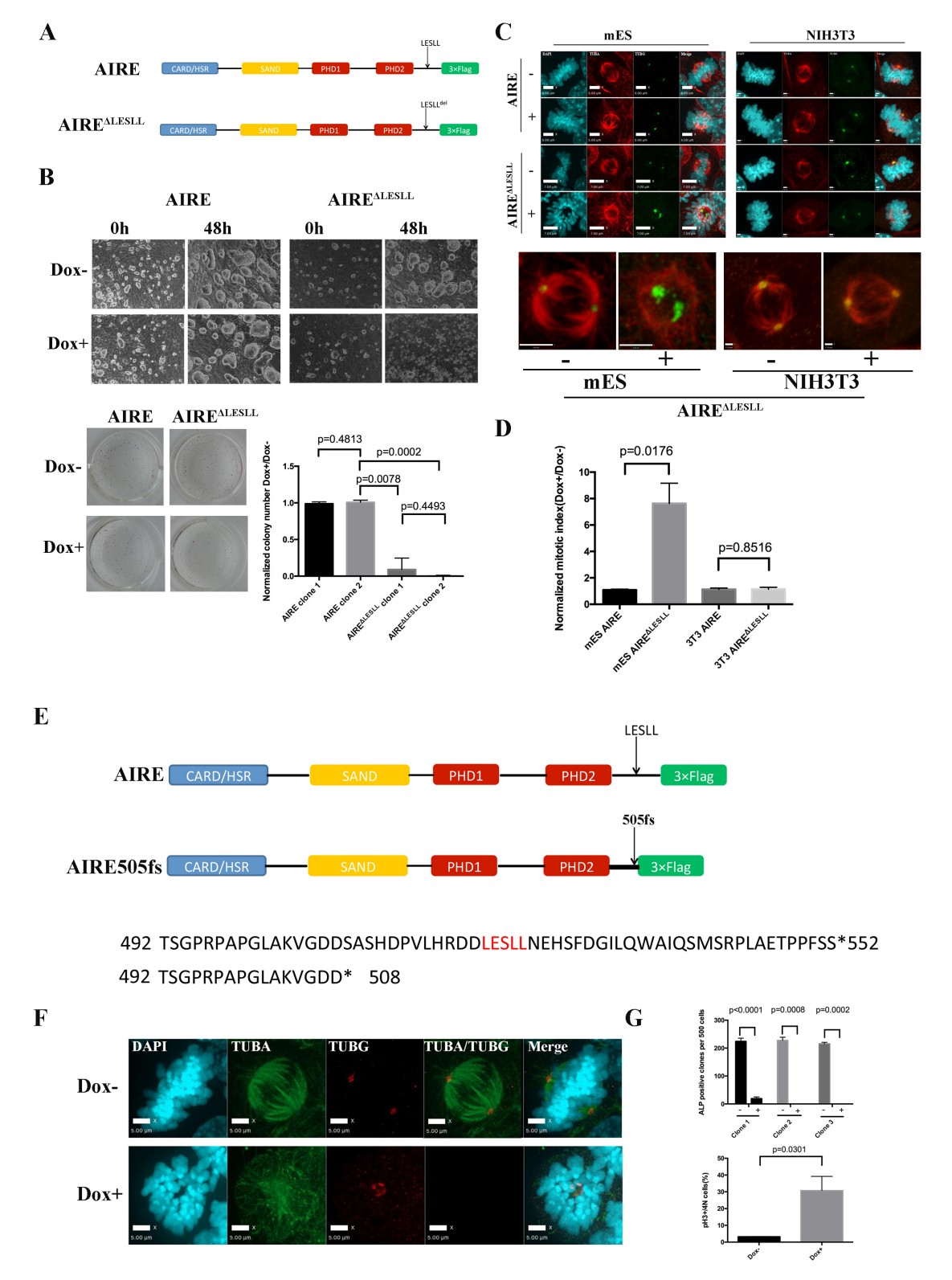

**Figure 4.** The last LxxLL(LESLL) motif is essential for the mitotic functions of Aire in ES cells. (**A**) Diagram showing Aire and Aire $^{\Delta LESLL}$ transgene structures. (**B**) Characterization of proliferation and colony formation of ES cells upon doxycycline-induced overexpression of AIRE or AIRE$^{\Delta LESLL}$. Representative bright field images of ES cell cultures 48 hr after doxycycline induction from three biological replicates (top panel). Representative images (bottom left panel) and quantification (bottom right panel) of ALP-stained colonies of AIRE or AIRE $^{\Delta LESLL}$-overexpressing ES cells 3 days after

*Figure 4 continued on next page*

*Figure 4 continued*

seeding. Quantification of ALP positive colonies was done from two independent clones for each transgene and normalized to Dox- controls. Data presented as mean ± sd of 3 biological replicates. *p*-values were calculated with Wilson's t-test. (C) Immunofluorescence staining of spindles in control (Dox-) and AIRE or AIRE $^{\Delta LESLL}$-overexpressing (Dox+) ES cells (left panel) and NIHT3T cells (right panel). Scale bar: 7 μm. Magnified images of spindles from AIRE $^{\Delta LESLL}$-overexpressing groups. Scale bar: 4 μm. (D) Mitotic index (number of cells in mitosis in Dox+/Dox- conditions) of AIRE and AIRE $^{\Delta LESLL}$-overexpressing ES cells and NIH3T3 cells. Number of cells in mitosis was determined by flow analysis for PI-4N/pH3 markers. Data presented as mean ± sd of 3 replicates. *p*-values were calculated with Wilson's t-test. (E) Diagram showing domain structure and partial amino acid sequence of human AIRE and AIRE505fs. (F) Immunofluorescence staining of spindles in control (Dox-) and AIRE505fs-overexperssing (Dox+) ES cells. Scale bar: 5 μm. (G) Characterization of colony formation and mitotic index of ES cells upon doxycycline-induced overexpression of Aire505fs. Top panel: number of ALP-positive colonies per well analyzed for 3 AIRE505fs transgenic ES cell clones without (Dox-) and with (Dox+) addition of doxycycline. Data presented as mean ± sd of 3 biological replicates for each clone, *p*-values were calculated with Wilson's t-test. Bottom Panel: percent of cells in mitosis in control (Dox-) and AIRE505fs overexpressing (Dox+) ES cells. Number of cells in mitosis was determined by flow analysis for PI-4N/pH3 markers (bottom panel). Data presented as mean ± sd of 3 replicates. *p*-values were calculated with Wilson's t-test.

The following figure supplements are available for figure 4:

**Figure supplement 1.** Both AIRE and AIRE $^{\Delta LESLL}$ localized to mitotic spindles in mES and NIH3T3 cells.

**Figure supplement 2.** AIRE $^{\Delta LESLL}$ activates the Spindle Assembly Checkpoint in ES cells.

overexpressing cells (*Figure 5—figure supplement 2*). These data suggested a crucial role of Aire in maintaining centrosome number and mitotic spindle pole integrity in ES cells.

## Aire is critical for spindle integrity in preimplantation embryos

To understand whether the mitotic functions of Aire in ES cells reflect its role in germ cells and the early embryo, we generated paternal *Aire* knockout mice by breeding *Aire$^{Fl/Fl}$* mice with *Alpl-Cre* transgenic mice and maternal *Aire* knockout mice by breeding *Aire$^{Fl/Fl}$* mice with *Zp3-Cre* transgenic mice, respectively. Both male germ cell-specific *Aire* knockout mice (*Aire$^{Fl/-}$;Alpl-Cre$^+$*) and female oocyte specific *Aire* knockout mice (*Aire$^{Fl/Fl}$;Zp3-Cre$^+$*) developed normally and were fertile (*Figure 6—figure supplement 1*), suggesting that, at least in the CD1 background, germ cell-specific depletion of Aire was not sufficient to induce consistent fertility defects. We then produced maternal/zygotic depleted (*m-z-*) embryos by breeding *Aire$^{Fl/-}$;Alpl-Cre$^+$* males and *Aire$^{Fl/Fl}$;Zp3-Cre$^+$*-females. Embryos were flushed at embryonic day (E) 3.5, the time when centrosome-dependent mitosis is established in the embryo (*Courtois et al., 2012*). As shown in *Figure 6A*, while control embryos were mostly at the blastocyst stage with a discernable blastocoel cavity, many of the *m-z-* embryos did not develop a blastocoel. Statistical analysis revealed that *m-z-* embryos showed a significantly lower blastocyst rate than controls (*Figure 6B*). Immunostaining revealed Aire protein in both inner cell mass (ICM) (Sox2+) and trophectoderm (Cdx2+) cells in control blastocysts (*Figure 6C*), but it was completely depleted in the *m-z-* blastocysts and morulae. *M-z-* embryos still expressed Sox2 and Cdx2 in a mutually exclusive pattern, indicating correct initiation of lineage specification (*Figure 6C* and *Figure 6—figure supplement 2*). However, we noted an overall decrease in total cell number in *Aire m-z-* embryos, indicative of a proliferation defect (*Figure 6B*). We then investigated the spindle structure in mitotic cells in *Aire m-z-* embryos and found an increased number of γ-tubulin (TUBG) positive foci compared to control cells, which formed abnormal, multipolar spindles (*Figure 6D and E*). These results suggested that, similar to ES cells, Aire also plays a critical role in sustaining centrosome number and spindle integrity in the cells of early embryos.

## Discussion

In this study, we report a novel function of the Autoimmune Regulator, *Aire*, in the mitotic processes of embryonic stem cells and the early embryo. Using the BioID method, we identified proximity partners for AIRE in ES cells. In addition to proteins functioning in general transcription and RNA processing, processes in which Aire was already known to function in somatic cells such as mTECs (*Abramson et al., 2010*; *Mathis and Benoist, 2009*), we identified many spindle associated proteins

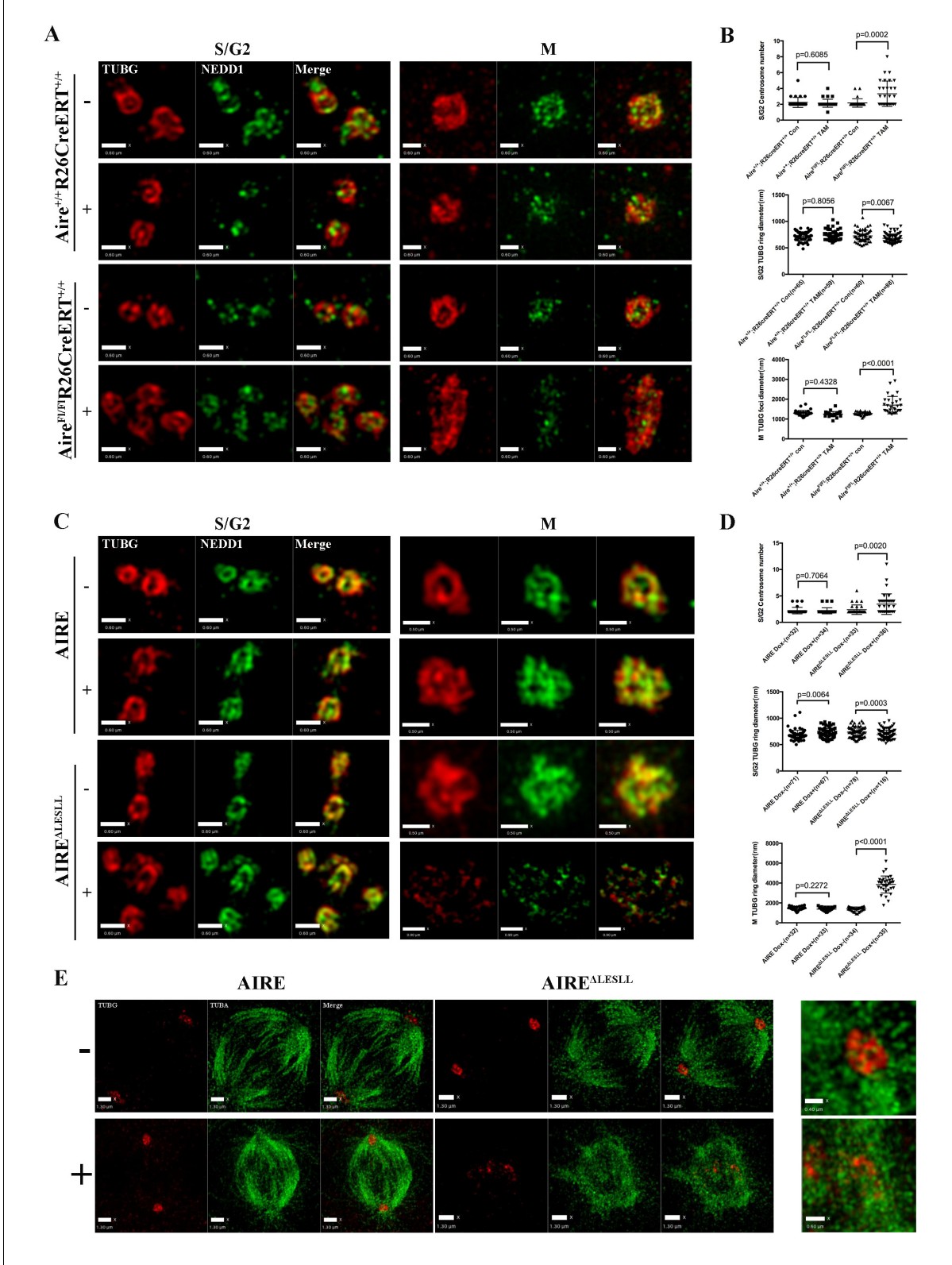

**Figure 5.** Aire is critical for centrosome number sustenance and the integrity of mitotic spindle poles. (**A**) Representative immunofluorescence SIM images of γ-tubulin (TUBG) and NEDD1 in control (Aire[+/+];R26CreERT[+/+] tamoxifen-/+and Aire[FL/FL];R26CreERT[+/+] tamoxifen-) and Aire[-/-] (Aire[FL/FL];R26CreERT[+/+] tamoxifen [+]) mES cells during S/G2 and M phases of the cell cycle. Scale bar: 0.7 μm. (**B**) Quantification of centrosome parameters in control and Aire[-/-] ES cells (groups same as in panel (**A**)). Centrosome number in 30 S/G2 cells (top panel) from three biological replicates were counted

*Figure 5 continued on next page*

Figure 5 continued

for each group, mean ± sd is shown. Each point represents one mitotic cell. *p*-values were calculated with Mann–Whitney–Wilcoxon test. γ-tubulin (TUBG) ring diameter in S/G2 cells (middle panel). Each point represents one centrosome. Mean ± sd was presented. *p*-values were calculated with Mann–Whitney–Wilcoxon test. γ-tubulin(TUBG) foci diameter in M phase cells (bottom panel). 30 spindle poles from three biological replicates were quantified for each group. Each point represents one spindle pole. Mean ± sd is presented. *p*-values were calculated with Mann–Whitney–Wilcoxon test. (C) Representative immunofluorescence SIM images of γ-tubulin (TUBG) and NEDD1 in control (Dox-) and AIRE or AIRE $^{\Delta LESLL}$-overexpressing mES cells (Dox+) during S/G2 and M phases of the cell cycle. Scale bar: S/G2 *Figure 0.6* μm; M images in top three panels 0.5 μm, and bottom panel 0.9 μm. (D) Quantification of centrosome parameters in AIRE or AIRE $^{\Delta LESLL}$-overexpressing mES cells (groups same as in panel (C)) Centrosome number in S/G2 cells (top panel). Each point represents one mitotic cell. Mean ± sd is presented. *p*-values were calculated with Mann–Whitney–Wilcoxon test. γ-tubulin (TUBG) ring diameter in S/G2 cells (middle panel). Each point represents one centrosome. Mean ± sd was presented. *p*-values were calculated with Mann–Whitney–Wilcoxon test. γ-tubulin(TUBG) diameter in M phase cells(bottom panel). Each point represents one spindle pole. Mean ± sd is presented. *p*-values were calculated with Mann–Whitney–Wilcoxon test. (E) Representative immunofluorescence SIM images of spindles and MOTCs (γ-tubulin and α-tubulin) in control (Dox-) and AIRE or AIRE $^{\Delta LESLL}$-overexpressing mES cells (Dox+) during M phase. Scale bar: 1.3 μm. Magnified view of MTOC in AIRE $^{\Delta LESLL}$ (Dox-) (top scale bar: 0.6 μm) and AIRE $^{\Delta LESLL}$ (Dox+) (bottom scale bar: 0.4 μm).

The following figure supplements are available for figure 5:

**Figure supplement 1.** Aire is critical for centrosome number sustenance and the integrity of mitotic spindle poles.

**Figure supplement 2.** AIRE $^{\Delta LESLL}$ overexpression caused defects in the pericentiole matter (PCM) around centrioles.

interacting with AIRE in ES cells. Indeed, spindle associated proteins were more enriched among AIRE-interacting proteins than were transcription-associated proteins. We then showed that AIRE localized to mitotic spindles in ES cells and determined that the HSR and SAND domains were responsible for spindle recruitment. The HSR domain has been proposed to be important for the self-dimerization of AIRE and for the localization of human *AIRE* to filamentous/microtubule cytoskeletal structures in somatic cell lines during interphase (*Rinderle et al., 1999*; *Pitkänen et al., 2001*; *Halonen et al., 2004*). However, mitotic spindle localization was not investigated in previous studies. The SAND domain is generally considered a DNA binding domain; however, in AIRE the critical amino acid residues for DNA binding are not conserved (*Purohit et al., 2005*; *Bottomley et al., 2001*; *Perniola and Musco, 2014*). Rather, our data suggest that the SAND domain plays a specific role in spindle localization of AIRE in ES cells.

Using conditional knockout strategies, we showed that *Aire*$^{-/-}$ ES cells and *m-z-* preimplantation embryo cells formed spindles with multiple centrosomes in mitosis, suggesting a critical function of *Aire* in centrosome sustenance and spindle assembly. To our knowledge this is a novel function for *Aire* and one that is likely to be ES cell/embryo specific since *Aire* is only expressed in a few restricted somatic cell types in adults and those cell types, such as mTECs, are generally not proliferative (*Gray et al., 2007*; *Hubert et al., 2008*; *Gardner et al., 2008*; *Yamano et al., 2015*). However, it is difficult at this point to exclude the possibility of *Aire* playing similar mitotic functions in rare populations of somatic stem cell types since the full range of expression of *Aire* is largely unknown.

We have previously shown that knocking down *Aire* in ES cells using shRNA caused proliferation and karyotype defects (*Gu et al., 2010*). Here we show that both proliferation and colony formation were impaired in *Aire*$^{-/-}$ ES cells. *Aire m-z-* preimplantation embryos also showed proliferation defects but early lineage segregation was not apparently disrupted. The fact that cell proliferation can still continue, albeit in an impaired form in Aire mutant ES cells and embryos, despite the clearly disturbed spindle and centrosome morphologies is not inconsistent with what is known about the roles of centrosomes in early embryonic cell cycles. It is known that cells can proliferate in the presence of multiple centrosomes by centrosome congregation (*Ring et al., 1982*; *Quintyne et al., 2005*), and in the absence of centrosomes, so long as a p53-mediated G1 checkpoint is somehow compromised (*Wong et al., 2015*; *Lambrus et al., 2015*). That checkpoint is likely compromised in ES cells and only begins to function at midgestation stages in mouse embryos (*Suvorova et al., 2016*; *Aladjem et al., 1998*; *Bazzi and Anderson, 2014*). In the preimplantation embryo, early mitotic divisions involve meiotic-like acentrosomal spindle assembly, while regular somatic-like mitosis where spindle poles are organized around centrosomes is only established around the blastocyst stage (*Courtois et al., 2012*).

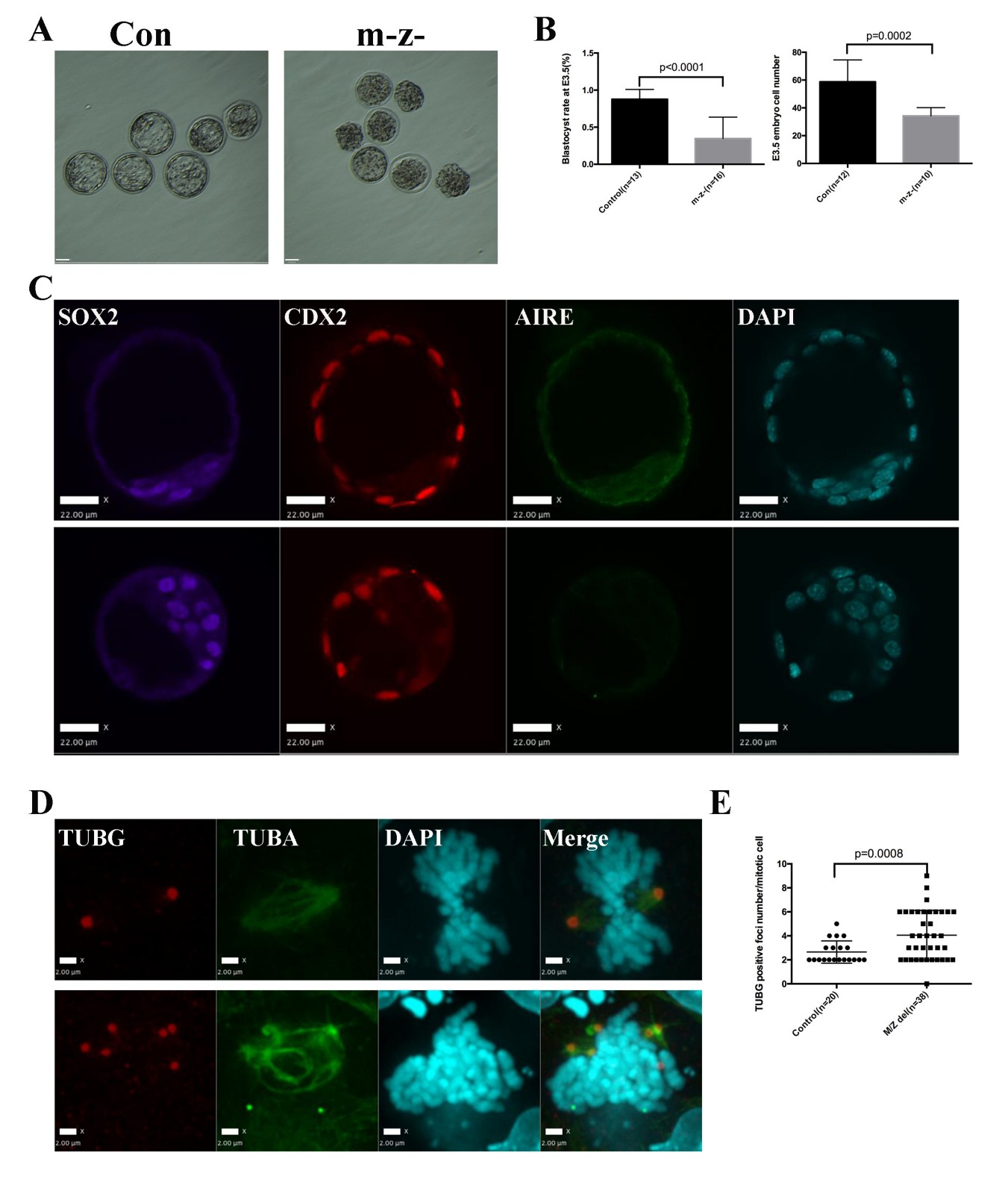

**Figure 6.** Aire m-/z- preimplantation embryos showed proliferation delay and spindle defects. (**A**) Brightfield images of control (Con) and Aire m-/z- embryos at embryonic day (**E**) 3.5. (**B**) Quantification of blastocyst-formation rate (left panel) and cell number at E3.5 (right panel) in control (Con) and Aire m-/z- embryos. Data is presented as mean ± sd. *p*-values were calculated with Wilson's t-test. (**C**) Immunofluorescence staining for Aire in control (Con) (upper) and Aire m-/z- (down) E3.5 embryos. Scale bar: 22 μm. (**D**) Representative immunofluorescence image of spindles (γ –tubulin (TUBG), α-

*Figure 6 continued on next page*

*Figure 6 continued*

tubulin (TUBA)) in E3.5 control (Con) (top) and Aire m-/z- (bottom) embryo cells. Scale bar: 2 µm. (E) Quantification of γ –tubulin (TUBG) positive foci per mitotic cell in control and Aire m-/z- embryos. Each point represents one mitotic cell. Mean ± sd is presented. *p*-values were calculated with Wilson's t-test.

The following figure supplements are available for figure 6:

**Figure supplement 1.** No obvious infertility phenotype was observed in paternal or maternal Aire knockout mice.

**Figure supplement 2.** Aire m⁻ z⁻ embryos didn't show obvious early lineage defects.

Both germ cell-specific *Aire* knockout males and females were fully fertile, suggesting that at least in the genetic background investigated (CD1) the depletion of Aire in germ cells was not sufficient to induce a consistent infertile phenotype, as had been reported before in *Aire*$^{-/-}$ conventional knockout mice (*Anderson et al., 2002*; *Ramsey et al., 2002*). It is plausible that the infertility defects reported in conventional *Aire* mutants may actually result from the combination of auto-immune and strain-dependent embryonic defects. Since human APECED patients also show high variability between individuals with regard to fertility (*Finnish-German APECED Consortium et al., 1997*; *Perheentupa, 2006*), the relationship between autoimmune defects, germline and embryo defects and genetic strain/background factors still remain to be dissected. It is not clear whether the proliferation delay and spindle defects observed in *Aire m-z-* blastocysts would result in later problems in development, given that both human and mouse embryos can overcome a certain level of mitotic and karyotypic defects (*Bolton et al., 2016*; *van Echten-Arends et al., 2011*).

We found the last LxxLL motif (LESLL) to be critical for the mitotic function of Aire in ES cells. Truncations involving this motif (AIRE$^{\Delta c70}$, AIRE $^{\Delta LESLL}$, AIRE505fs) all perturbed centrosome numbers and spindle pole integrity when overexpressed in ES cells. Interestingly, the effect seemed to be specific to ES cells since the overexpression of the same AIRE $^{\Delta LESLL}$ in NIH3T3 cells did not cause mitotic defects. These data support the idea that Aire plays a specific role in mitotic processes in ES cells relative to somatic cells Interestingly, not many spindle-associated proteins were identified as interaction partners of Aire in previous studies using somatic cell lines overexpressing Aire (*Abramson et al., 2010*; *Meloni et al., 2010*; *Waterfield et al., 2014*). This could also support an ES cell-specific mitotic function for Aire but could also be related to technical differences between the BioID method used in this study and the co-immunoprecipitation or yeast two-hybrid methods used by previous studies (*Lambert et al., 2015*).

The mitosis-disrupting effect of overexpressing AIRE $^{\Delta LESLL}$ was more severe than the phenotype following Aire depletion in ES cells. The reasons for this are not entirely clear. However, our BioID data suggest that Aire may act in a multi-protein complex, including SPICE1 (*Archinti et al., 2010*), Human Augmin Complex (HAUS) complex proteins (Haus5 and Haus8) and CLASP proteins (CLASP1 and CLASP2) (*Lawo et al., 2009*; *Bratman and Chang, 2008*), which have all been reported to function in centrosome maturation/duplication and mitotic spindle organization. Removal of AIRE from the complex by gene knockout could result in a partially functional complex, while replacement with a dominant negative form like AIRE $^{\Delta LESLL}$ could disrupt the conformation of that complex and cause a more complete loss of function, as have been shown before in other experimental systems (*Veitia, 2007*; *Papp et al., 2003*). Another possible explanation for the observation of more severe mitotic phenotype in AIRE $^{\Delta LESLL}$ overexpressing cells than Aire depleted cells is the existence of redundant pathways. To this end, a particularly interesting gene is Deaf1, another SAND domain containing transcription regulator structurally related to Aire that conducts similar functions to activate expression of auto-antigen genes in the peripheral lymphoid system (*Yip et al., 2009*). Interestingly, Deaf1 has been shown to be expressed in embryos and ES cells and maternal zygotic depletion of Deaf1 homolog in Drosophila embryos caused early embryo arrest (*Park et al., 2013*; *Gu et al., 2010*; *Veraksa et al., 2002*). It would be interesting to study the effect of depletion of Deaf1 or combined depletion of Aire and Deaf1 to understand the function of this group of genes in mES cells. It is also important to point out that a previously reported disease causing mutation (AIR-E505fs) in humans (*Ishii et al., 2000*) results in the deletion of the LESLL motif. Mouse AIRE proteins

with an analogous truncation produced a mitosis-disrupting phenotype similar to AIRE $^{\Delta LESLL}$ in ES cells. This suggests that some patient mutations might have mitosis-disrupting effects in early embryos, which might be related to the reported fertility phenotypes.

The mechanism by which Aire sustained the centrosome number stability and structure integrity in mES cells is not clear at this point. However, the acquisition of multiple centrosomes, especially in the situation of AIRE $^{\Delta LESLL}$ overexpression, occurred within 12 hr of doxycycline induction, which is less than one cell cycle of mES cells. This suggests excess duplication of centrosomes. CDK2 has been identified as a proximity partner of AIRE in mES cells. Elevated centrosome associated CDK2 activity in a p53 deficient cellular environment has been shown to cause centrosome over-duplication (*Tarapore and Fukasawa, 2002*; *Adon et al., 2010*). Interestingly, mES cells possess sustained elevated CDK2 activity and have inefficient p53 functions but normally manage to maintain normal centrosome numbers (*Stead et al., 2002*; *Aladjem et al., 1998*). We speculate that, by interacting with CDK2, AIRE somehow limits the centrosome over-duplication that would be activated by the sustained elevated CDK2 activity. Furthermore AIRE also interacts with a group of centrosome associated proteins including SPICE1 (*Archinti et al., 2010*) and Human Augmin Complex (HAUS) complex proteins (Haus5 and Haus8) (*Lawo et al., 2009*) which have been shown to function in maintenance of centrosome number and integrity. Disruption of this interaction could also contribute to the centrosome phenotypes of Aire depletion and AIRE $^{\Delta LESLL}$ overexpression in mES cells. However, since AIRE also interacted with proteins in mES cells that function in processes such as transcription regulation and RNA processing, it is still possible that defects in those processes upon Aire depletion or AIRE $^{\Delta LESLL}$ overexpression indirectly contributed to the mitotic phenotype. Further study is required to separate out the different functions of Aire in mES cells and embryos.

Aside from the implications in mitosis regulation in ES cells and early embryos, this study also raises an interesting speculation concerning the evolution of the immune system. As a relatively newly evolved system, the immune system frequently co-opts proteins from basic processes like cell proliferation and embryonic development. For example, the Toll receptor plays a role in the formation of the dorsal-ventral axis in *Drosophila* (*Valanne et al., 2011*) and the Toll-like receptors not only form the basis of innate immunity, but have also been shown to play a role in proliferation in cell types including ES cells (*Taylor et al., 2010*). Activation-induced cytidine deaminase (AID), a critical enzyme for somatic hypermutation, gene conversion, and class-switch recombination in B cells (*Peled et al., 2008*), has also been shown to function in epigenetic reprogramming of primordial germ cells and induced pluripotent stem cells (*Bhutani et al., 2010*). The possibility of a similar, multi-functional role of Aire is especially intriguing because a central tolerance system coordinated by activating promiscuous gene expression is not required until the evolution of the adaptive immune system and its diversification of immune receptors (*Schatz and Swanson, 2011*). Organisms may have adapted molecules like Aire from existing biological processes to establish a tolerance mechanism and cope with the random immune receptor diversification. It would be an interesting question to trace the molecular evolution of Aire and determine its function in embryo development in lower organisms.

In conclusion, this study uncovered a novel role of Aire in regulating mitosis in ES cells with implications for the biology of proliferation regulation in ES cells and early embryos.

## Materials and methods

Sources of materials and DNA sequences are shown in *supplementary files 1* and *2*.

### Constructs

To generate the C-terminal Aire-3XFlag-tag expression construct, the full length coding sequence (CDS) of Aire transcription variant 1(longest variant) was amplified from mES cDNA and cloned into the pCMV-3Xflag 14 vector ((Sigma, Oakville, Canada)). To generate a doxycycline (Dox)-inducible Aire-3XFlag-tag expression construct, Aire-3XFlag-tag cDNA was cloned into the pDonor221 vector (Invitrogen, Thermo Fisher Scientific, Waltham, MA) and then a Dox-inducible Piggybac destination vector (PB-T-RfA, kindly provided by Dr. Andras Nagy) using a Gateway Cloning Kit (*Li et al., 2013*) (Thermo Fisher Scientific). To generate a C-terminal Aire-BirA*-HA expression construct, Aire CDS was cloned into the pcDNA3.1 MCS-BirA(R118G)-HA vector(a

kind gift from Dr. Kyle Roux) (Addgene Cambridge, MA) (*Roux et al., 2012*). To generate a Dox-inducible C-terminal Aire-BirA*-HA expression construct, Aire-BirA*-HA was cloned into a Dox-inducible Piggybac destination vector (PB-T-RfA) using Gateway cloning, as described before. A Dox-inducible mCherry-BirA*-HA expression plasmid was constructed similarly by fusing mCherry CDS to BirA*-HA. Mutations of Aire were introduced into the pDonor221 Aire-3XFlag-tag vector through PCR mediated mutagenesis using InFusion cloning kit (Takara Bio, USA) and then cloned into the PB-T-RfA destination vector.

For CRISPR experiments, the Cas9 CDS was PCR amplified from the pX330-U6-Chimeric_BB-CBh-hSpCas9(pX330) plasmid (a kind gift from Dr. Feng Zhang) (Addgene) (*Cong et al., 2013*) and inserted into the pCS2 +plasmid using BamHI:EcoRI sites. Guide RNAs (gRNAs) were designed using the Optimized CRISPR design tool(http://crispr.mit.edu). DNA oligos coding sgRNAs were synthesized (Sigma) and inserted into the pX330 plasmid in BbsI site (*Cong et al., 2013*).

All the primer sequences are shown in *supplementary file 2*. Plasmids will be made available through Addgene.

## In vitro transcription

To produce Cas9 mRNA, the pCS2+-Cas9 plasmid was linearized with NotI restriction digestion and used as template to in vitro transcribe mRNAs using mMESSAGE mMACHINE SP6 Transcription Kit (Thermo Fisher Scientific). For the production of sgRNAs, sgRNA coding sequences were PCR-amplified from the pX330 plasmids with primers containing T7 promoters (*supplementary file 2*) and used as template to produce sgRNA using the MEGAshortscript T7 Transcription Kit (Invitrogen). All RNA products were purified using RNeasy Mini Kit (QIAGEN,Toronto. ON. Canada) following the cleanup protocol.

## Superovulation

Female mice of 5–8 weeks age were injected with 5 IU each pregnant mare serum gonadotropin (PMSG) (Sigma, Oakville, Canada) and human chorionic gonadotropin (hCG) (Sigma, Oakville, Canada), 48 hr apart. The females were then mated with males of 8–10 weeks age. Vaginal plugs were checked the following morning and plugged female were counted as 0.5dpc.

## Microinjection

Superovulated females were acquired from the Transgenic Core of The Centre for Phenogenomics and mated with CD1 males. Zygotes were isolated from the oviducts of 0.5dpc females and the cumulus cells were removed by incubating in 50 µg/ml Hyaluronidase (EMD Millipore) at room temperature for five mins and kept in KSOM. Pronuclear injections were performed using a Leica microscope and micromanipulators (Leica Microsystems Inc., Richmond Hill, Canada). The injection pressure was supplied by a FemtoJet (Eppendorf). The injections were performed in a drop of EmbryoMax M2 Medium (EMD Millipore) covered with mineral oil (Zenith Biotech, Guilford, CT).

## Mouse lines

All animal work was carried out following Canadian Council on Animal Care Guidelines for Use of Animals in Research and Laboratory Animal Care under protocols approved by The Centre for Phenogenomics Animal Care Committee (protocol number: 20–0026 hr). All mouse lines were generated on a CD1 (breeding stock from Charles River, Montreal, Canada) background. *Aire-3xFlag-tag* (CD1(ICR)-*Aire^tm2(Flag-Aire)Jrt*) and *Aire^Fl/Fl*(CD1(ICR)-*Aire^<tm2Jrt>*)mice were generated using CRISPR-Cas9 genome editing (*Yang et al., 2014*). *Aire-3xFlag* tag mice were generated by microinjecting the pronucleus of zygotes with a mixture of 20 ng/ul Cas9 mRNA, 10 ng/ul sgRNA (targeting the terminal codon region of *Aire*) and 10 ng/ul single strand ultramer oligo donor (ssODN) (Integrated DNA Technologies(IDT) Coralville, Iowa, USA) encoding 3XFlag-tag and 80 bp homology arm on both 5′ and 3′ sides. Injected zygotes were cultured overnight in KSOM (EMD Millipore, Etobicoke ON. CA.) with 50 uM SCR7 (Xcess Biosciences, San Diego CA. USA.) to facilitate homology directed repair (HDR) (*Maruyama et al., 2015*) and transferred to pseudo-pregnant females the following day. Founder mice were genotyped (see *supplementary file 2* for primer sequences) and the PCR-amplified targeting region was Sanger sequenced to validate

the insertion. Founder mice were out-crossed to CD1 mice for four generations before used to establish ES cell lines. The $Aire^{Fl/Fl}$ allele was designed as two loxp sites inserted into intron 7 and intron 9 of *Aire*, flanking critical exons 8 and 9. The removal of exon 8 and 9 by Cre-mediated recombination caused a frame shift and premature termination of all known coding variants of *Aire*. The mouse line was derived by a two-step procedure. The pronucleus of zygotes was microinjected with *Cas9* mRNA, sgRNA targeting intron seven and ssODN encoding a Loxp site with 80 bp homology arms on both 5' and 3' sides. The embryos were cultured and transferred same as before. Founder mice were genotyped and sequence-validated. Mice carrying one loxp site were bred to homozygosity, and a similar round of CRISPR-Cas9 editing was performed on zygotes from this line to target the second loxp into intron 9. The $Aire^{Fl/Fl}$ ($ICR$-$Aire^{tm2Jrt}$) mice were then bred for three generations to CD1, followed by crossing to *R26CreERT* mice (B6;129-Gt(ROSA)26Sortm1$^{(cre/ERT)Nat}$/J) to make $Aire^{Fl/Fl}$ *R26CreERT* mice (*Badea et al., 2003*). The $Aire^{Fl/Fl}$ mice were also crossed to *Alpl-Cre*(129-$Alpl^{tm1(cre)Nagy}$/J) mice to generate primordial germ cell-specific *Aire* knockout mice (*Lomelí et al., 2000*). The $Aire^{Fl/Fl}$ mice were also crossed to *Zp3-Cre* (C57BL6-Tg(*Zp3-cre*)93Knw/J) mice to generate oocyte-specific *Aire* knockout mice (*de Vries et al., 2000*).

## Cell lines derivation and culture

The AmES8 cell line was a wildtype mES cell line derived from blastocysts acquired from breeding of heterozygotes of B6.129S2-*Aire*$^{tm1.1Doi}$/J mice (*Anderson et al., 2002*). The cell line was derived and maintained in 2i-LIF + Serum conditions (1:1 mixing of LIF-Serum Medium and 2i-LIF medium (*Ying et al., 2008*) on mitomycin C(MMC) (Sigma) inactivated E15.5 mouse embryonic fibroblast (MEF) feeders. The pluripotency of the cell line has been tested by in vitro differentiation and chimera formation. Flag-Aire mES cell lines were derived from blastocysts acquired from homozygote breeding of Aire-3xFlag-tag mice and maintained in 2i-LIF-Serum condition. The $Aire^{+/+}$; $R26CreERT^{+/+}$, $Aire^{Fl/Fl}$;$R26CreERT^{+/+}$, $Aire^{Fl/+}$;$R26CreERT^{\pm}$ and $Aire^{Fl/Fl}$;$R26CreERT^{\pm}$ mES cell lines were derived from blastocysts acquired from breeding $Aire^{Fl/+}$;$R26CreERT^{\pm}$ mice and maintained in 2i-LIF + Serum conditions.

For 4-Hydroxytamoxifen(TAM) (Sigma) treatment, TAM concentration was first titrated on $Aire^{+/+}$; $R26CreERT^{+/+}$, and $Aire^{Fl/+}$;$R26CreERT^{\pm}$ mES cells respectively to exclude proliferation toxicity to Cre only cells. Based on the titration, $R26CreERT^{+/+}$ and $R26CreERT^{\pm}$ cells were treated with 5 ng/ml and 20 ng/ml TAM in 2i-LIF + Serum medium for 24 hr respectively and then maintained in 2i-LIF-Serum medium. The treated cells were then passaged and used for experiments.

To derive mES cell lines carrying Dox inducible transgenes (TetON cell lines), AmES8 cells were co-transfected with PBASE (encoding piggyback transposase), PBCA-rtTA and respective destination vectors as previously described (*McDonald et al., 2014*) and selected for one week with 2 µg/ml Puromycin (Sigma). The transfected cells were then either single cell-sorted into 96 well plates or plated sparsely on 10 cm dishes with feeders for clone picking. Clonal cell lines were acquired and tested for inducible expression by immunofluorescence. For the Dox inducible expression experiments, cells were cultured in 1 µ g/ml Dox (Sigma) for various times (Figure legends) and then cells were collected for downstream experiments.

NIH3T3 cells (*Jainchill et al., 1969*) were acquired from ATCC and cultured in DMEM medium (Thermo Fisher Scientific) supplemented with 10% FBS(Thermo Fisher Scientific). The TetON cell lines were established similarly to mES cells.

All the cell lines were tested for Mycoplasma using the Universal Mycoplasma Detection Kit (ATCC, Manassas, VA. USA) and were negative and authentificated using STR method by IDEXX Bioresearch to be mouse origin.

## Proximity biotinylation (BioID)

Transgenic AmES8 cells with Dox inducible mCherry-BirA*-HA or Aire-BirA*-HA were grown on gelatin-coated tissue culture flasks in 2i-LIF-Serum. For BioID, the transgenes were first induced for 24 hr with 1 µg/ml Dox, then biotin was added to the medium to a final concentration of 50 µM for 24 hr. The streptavidin purification and protein digestion with trypsin were performed as previously described (*Lambert et al., 2015*). Samples were collected from three independent batch of experiments and assigned as three biological replicates.

## LC-MS/MS data acquisition

A spray tip was formed on fused silica capillary column (0.75 μm ID, 350 μm OD) using a laser puller (program = 4; heat = 280, FIL = 0, VEL = 18, DEL = 200). 10 cm (±1 cm) of C18 reversed-phase material (Reprosil-Pur 120 C18-AQ, 3 μm) was packed in the column by pressure bomb (in MeOH). The column was pre-equilibrated in buffer A (6 μL) before being connected in-line to a NanoLCUltra 2D plus HPLC system (Eksigent) coupled to a LTQ-Orbitrap Elite (Thermo Fisher Scientific) equipped with a nanoelectrospray ion source (Proxeon Biosystems, Thermo Fisher Scientific). The LTQ-Orbitrap Elite instrument under Xcalibur 2.0 was operated in the data dependent mode to automatically switch between MS and up to 10 subsequent MS/MS acquisitions. Buffer A was 99.9% H2O, 0.1% formic acid; buffer B was 99.9% acetonitrile, 0.1% formic acid. The HPLC gradient program delivered an acetonitrile gradient over 125 min. For the first twenty minutes, the flow rate was 400 μL/min at 2% B. The flow rate was then reduced to 200 μL/min and the fraction of solvent B increased in a linear fashion to 35% until 95.5 min. Solvent B was then increased to 80% over 5 min and maintained at that level until 107 min. The mobile phase was then reduced to 2% B until the end of the run (125 min). The parameters for data dependent acquisition on the mass spectrometer were: one centroid MS (mass range 400–2000) followed by MS/MS on the 10 most abundant ions. General parameters were: activation type = CID, isolation width = 1 m/z, normalized collision energy = 35, activation Q = 0.25, activation time = 10 msec. For data dependent acquisition, the minimum threshold was 500, the repeat count = 1, repeat duration = 30 s, exclusion size list = 500, exclusion duration = 30 s, exclusion mass width (by mass)=low 0.03, high 0.03.

## Mass spectrometry data analysis

RAW mass spectrometry files were converted to mzXML using ProteoWizard (3.0.4468; [Kessner et al., 2008]) and analyzed using the iProphet pipeline (Shteynberg et al., 2011) implemented within ProHits (Liu et al., 2016) as follows. The database consisted of the mouse RefSeq protein database (version 53) supplemented with 'common contaminants' from the Max Planck Institute (http://141.61.102.106:8080/share.cgi?ssid = 0f2 gfuB) and the Global Proteome Machine (GPM; http://www.thegpm.org/crap/index.html). The search database consisted of forward and reversed sequences (labeled 'DECOY'); in total 58202 entries were searched. The search engines used were Mascot (2.3.02; Matrix Science) and Comet (2012.01 rev.3; [Eng et al., 2015]), with trypsin specificity (two missed cleavages were allowed) and deamidation (NQ) and oxidation (M) as variable modifications. Charges + 2,+3 and+4 were allowed and the parent mass tolerance was set at 15 ppm while the fragment bin tolerance was set at 0.6 amu. The resulting Comet and Mascot search results were individually processed by PeptideProphet (Keller et al., 2002), and peptides were assembled into proteins using parsimony rules first described in ProteinProphet (Nesvizhskii et al., 2003) into a final iProphet protein output using the Trans-Proteomic Pipeline (TPP; Linux version, v0.0 Development trunk rev 0, Build 201303061711). TPP options were as follows. General options are -p0.05 -x20 -PPM -d"DECOY', iProphet options are –ipPRIME and PeptideProphet options are –OpdP. All proteins with a minimal iProphet protein probability of 0.05 were parsed to the relational module of ProHits. Note that for analysis with SAINT, only proteins with iProphet protein probability ≥0.95 are considered. This corresponds to an estimated protein-level FDR of ~0.5%.

## Interaction proteomics scoring

SAINTexpress analysis (Teo et al., 2014) was performed on the iProphet results (filtered at >0.95 protein probability) as follows. SAINTexpress (v 3.3) was used to estimate the probability of proximity interaction in the individual biological triplicates for Aire and mCherry control. Averaged probabilities (AvgP) were used to estimate the Bayesian false discovery rates. Hits with ≤1% FDR were deemed 'high confidence'. Downloadable files and all raw mass spectrometry files are deposited in the MassIVE repository housed at the Center for Computational Mass Spectrometry at UCSD (http://proteomics.ucsd.edu/ProteoSAFe/datasets.jsp). The datasets has been assigned the MassIVE IDs MSV000080386 (ftp://MSV000080386@massive.ucsd.edu). The dataset was assigned the ProteomeXchange Consortium (http://proteomecentral.proteomexchange.org) identifiers PXD005529. The dataset is currently password protected until publication; the password is Aire.

To further improve the specificity of the interactors detected by BioID following a SAINT analysis, we compared the results obtained here with our previous effort (*Lambert et al., 2015*). We found that Pccb, Pcca,, Mccc2, Acaca, Acacb and Mccc1 were detected at high level (more than 20 spectral count in nine distinct control purifications) and thus removed them from our final interactors list.

## Gene ontology(GO) analysis

Gene ontology analysis was performed with DAVID server with default parameters (*Huang et al., 2009b*, *2009a*). Data were presented as p values for Biological Process(BP_DIRECT) and Cellular Compartment(CC_DIRECT)

Western Blotting:

Western blotting was performed as previously described. (*Gu et al., 2010*). Samples were normalized by cell numbers.

## Cell growth assay

mES cells were seeded at a density of $10^5$ cells/well on gelatin coated six well tissue culture plates (Nunc) in growth medium and cultured for 48 hr (Passage 1(P1)). Cell numbers were counted at this point with a hemocytometer and recorded as P1 cell number. The P1 cells were subcultured at $10^5$ cells/well on gelatin coated six well tissue culture plates (Nunc) in growth medium again and cultured for a further 48 hr (Passage 2(P2)). Cell numbers were counted at this point with a hemocytometer and recorded as P2 cell number.

## Colony formation assay

Colony formation assays were performed as previously described (*Gu et al., 2010*). Briefly, mES cells were seeded at a density of 500 cell/well in 24 well tissue culture plates on MMC inactivated E15.5 MEFs (Nunc) in growth medium. ALP staining was performed after 72 hr with the Leukocyte Alkaline Phosphatase(ALP) Kit (Sigma) and ALP positive colonies were counted manually under a dissection microscope.

## Immunofluorescence

Cells for immunofluorescence were grown on gelatin-coated circular coverslips (Thickness 1.5) (Thermo Fisher Scientific). For all stainings, with the exception of staining for centriole acetyl-tubulin (see below), cells were fixed with 4% paraformaldehyde (PFA) for 20 min at room temperature. They were then permeabilized with Dulbecco's phosphate-buffered saline (DPBS) (Thermo Fisher Scientific) containing 0.5% TritonX-100 for 15 min and then blocked in DPBS containing 0.1% TritonX-100% and 5% FBS (Blocking Buffer(BB) for 1 hr at room temperature. They were then incubated with primary antibodies (*supplementary file 1*) diluted in BB overnight at 4°C. After thorough washing, the cells were incubated with fluorophore-conjugated secondary antibodies (*supplementary file 1*) for 45 min at room temperature. They were then washed and mounted on glass slides with the VECTASHIELD HardSet Antifade Mounting Medium with DAPI (Vector) and kept in 4°C before imaging. For cold treatment for analyzing acetyl-tubulins, cells were removed from the incubator and left on ice for 60 min before fixed with pre-chilled 100% methanol for 10 min at −20°C. The cells were then blocked with blocking buffer and stained and mounted similarly. The samples were then analyzed by either a spinning disk confocal microscopy (Quorum) or a Structured Illumination Microscope (Zeiss).

## Proximity ligation amplification(PLA) Assay

PLA assay is a immuno-PCR based assay to visualize proximity relationship of a pair of proteins in situ. (*Söderberg et al., 2006*) PLA assay was performed using Duolink In Situ Orange Starter Kit Mouse/Rabbit (Sigma) according to the product manual. All the PLAs were performed in TetON-3XFlag-Aire mES cells after 12 hr induction with 1µ g/ml Dox. Uninduced(Dox⁻) cells were used as negative control and the PLA between M2 anti-Flag mouse monoclonal antibody(Sigma) and corresponding rabbit antibodies against AIRE's interacting partners were detected with PLA kit and visualized using a spinning-disc confocal microscope(Quorum).

## Confocal microscopy analysis

Images were acquired using a Zeiss Axiovert 200 inverted microscope equipped with a Quorum spinning disk confocal scan head, a Hamamatsu C9100-13 EM-CCD camera, and Velocity(version 6.3.1) aquisition software. Spindle images of both cultured cells and early embryos were acquired as Z-stacks (at 0.25 μm intervals) with a 63x oil (NA = 1.35) objective. PLA images and images of pluri-potency markers in mES cells were acquired as Z-stacks (at 0.5 μm intervals) with an 20x air (NA = 0.75) objective. Images of lineage markers in early embryos were were acquired as Z-stacks (at 2 μm intervals) with an 20x air (NA = 0.75) objective. Images were visualized and analysed using Velocity (version 6.3.1).

## Structured illumination microscope (SIM)

Images were acquired using the structured illumination module of the Zeiss Elyra PS1. A 63x/1.4 objective, in combination with a 1.6x optovar, was used for data acquisition. Images spanning the centrosomes or spindles were taken as Z-stacks with 0.110 μm optical sections. The final SIM images were reconstructed and aligned using the automatic processing toolbox of Zeiss Zen with 3D-SIM mode. The SIM Images were then visualized by Velocity 6.3 software and presented as extended focus views.

## Quantification of mitotic γ-tubulin (TUBG) foci and centrosomes

For quantifications of mitotic γ-tubulin foci in confocal images, metaphase cells were identified by chromosome morphology under DAPI channel and then imaged for α-tubulin and γ-tubulin. The Images were then visualized with Velocity 6.3 software as extended focus views and γ-tubulin foci were manually counted.

For quantification of S/G2 centrosomes in SIM Images, S/G2 centrosomes were identified as clusters of 2 or multiple γ-tubulin positive foci under wide field visualization and Z-Stack SIM Images acquired and reconstructed by 3D-SIM. The images were then visualized with Velocity 6.3 software as extended focus views and centrosome numbers were manually counted in the images. The diameter of the γ-tubulin rings was measured and calculated as the average of length of the long and short axis of the rings in SIM images measured with Velocity. For quantification of mitotic spindle poles, mitotic metaphase cells were identified by chromosome morphology under DAPI channel and Z-Stack SIM images were then acquired and reconstructed by 3D-SIM. The diameter of TUBG foci was measured and calculated as the average of length of the long and short axis of the γ-tubulin positive regions.

## Flow cytometry

Intracellular staining flow cytometry analyses were performed according to protocol from Cell Signaling Technology (https://www.cellsignal.com/common/content/content.jsp?id=flow). Briefly, cells were trypsinized to single cells and then fixed with 4% PFA for 10 min at 37°C. They were then permeabilized with pre-chilled (−20°C) 90% methanol for 1 hr on ice. After three washes in flow buffer (DPBS containing 1% BSA), the cells were incubated in Pacific blue conjugated pS10-H3 antibodies(Cell Signaling Technology) diluted in flow buffer in the dark for 1 hr at room temperature. After three washes in flow buffer (DPBS containing 1% BSA), the cells were incubated in Propidium Iodide (PI)/RNase Staining Solution (Cell Signaling Technology) in the dark for 30 min at 37°C and then stored at 4°C before analysis (cells could be stored for up to 1 week). The cells were then analyzed on a BD LSRII-CFI BGRV and the data were analyzed by Flowjo software (FlowJo, LLC).

## Realtime PCR

Total RNA was isolated with Trizol (Invitrogen) and cDNAs were synthesized with QuantiTect Reverse Transcription Kit (Qiagen). Realtime PCR was performed using LightCycler 480 SYBR Green I Master Mix (Roch) with primers listed in (*Supplementary file 2*) The expression levels were calculated with $\Delta\Delta$CT methods.

## Fertility test

For the fertility test of paternal *Aire* knockout mice (patΔ), 2 pairs of *Aire*$^{+/-}$;*Alpl-Cre*$^+$ x CD1 breeding pairs were set up as control and 2 pairs of *Aire*$^{+/-}$;*Alpl-Cre*$^+$XCD1 breeding pairs were set up as *Aire*$^{patΔ}$. For the fertility test of maternal Aire knockout mice (matΔ), 3 pairs of CD1X*Aire*$^{Fl/Fl}$;*Zp3-Cre*$^-$ breeding pairs were set up as control while 3 pairs of CD1X*Aire*$^{Fl/Fl}$;*Zp3-Cre+* breeding pairs were set up as *Aire*$^{matΔ}$. All the breedings were started at 8 weeks age and monitored for 3 months. The litter number and litter size were recorded. The pups were monitored till weaning (21 days) and no obvious defects were observed.

## Embryo collection and immunostaining

For the analysis of *m-z-* embryos, control embryos were generated by breeding Aire$^{Fl/+}$;*Alpl*-Cre$^-$ males and superovulated Aire$^{Fl/Fl}$;*Zp3-Cre*$^-$ females and *m-z-* embryos were generated by breeding Aire$^{Fl/-}$;*Alpl*-Cre$^+$ males and superovulated *Aire*$^{Fl/Fl}$;*Zp3-Cre*$^+$ females. Embryos were flushed from uterus at E3.5 and fixed in 4% PFA and imaged under a stereoscope for blastocyst rate analysis. Embryos derived from each female were considered as one experimental unit for blastocyst rate analysis. The embryos were then permeabilized with Dulbecco's phosphate-buffered saline (DPBS) containing 0.5% TritonX-100 for 15 min and blocked in DPBS containing 0.1% TritonX-100% and 5% FBS (Blocking Buffer(BB) for 1 hr at room temperature. After removing the zona pellucida with EmbryoMax Acidic Tyrode's Solution (Millipore), the embryos were incubated with primary antibodies overnight at 4°C as indicated in Figures and Figure legends. The next day, embryos were washed in BB three times and stained with fluorophore-conjugated secondary antibodies (*Supplementary file 1*) for 45 min at room temperature. The embryos were then mounted in VECTASHIELD HardSet Antifade Mounting Medium with DAPI (Vector) and imaged under a spinning disk confocal microscope (Quorum) and analyzesd as stated in Confocal Analysis section.

## Statistical analysis

No statistical methods were used to pre-determine sample size. For cellular expriments, 2–3 biological replicates were performed. For 4-OH-tamoxyfen or Doxycycline treatment experiment, independent sets of control and treated cells were considered as biological replicates. For mouse fertility test, 2–3 breeding pairs of corresponding genotypes were analyzed. For blastocyst rate analysis, embryos isolated from 10 to 20 females from each groups were analyzed. For blastocyst cell number analysis, embryos collected from at least three females were analyzed in each group. For spindle analysis in blastocysts, 20–30 mitotic cells from at least 10 blastocysts of each genotypes were analyzed. The experiments were not blinded from investigators: the identities of the samples were known throughout the experiment. All statistics were performed with Prism software(GraphPad Software, Inc. La Jolla, CA. USA). For cell growth, colony formation and qPCR, Wilson's t-test was performed. For other data, data were first tested with D'Agostino and Pearson omnibus normality test, then Wilson's t-test or Mann–Whitney–Wilcoxon test were used for data that passed or did not pass the normality test respectively.

## Acknowledgements

This work was funded by Canadian Institutes of Health Research (CHIR grant #FDN-143334) (JR). BG was supported by Ontario Institute of Regeneration Medicine (OIRM) Postdoc Fellowship. A-CG is the Canada Research Chair in Functional Proteomics and the Lea Reichmann Chair in Cancer Proteomics and was supported by the Canadian Institutes of Health Research (FDN143301). J.PL was supported by a Cancer Research Society Scholarship for the Next Generation of Scientists. We thank Dr. Andras Nagy and Claudio Monetti from the Lunenfeld-Tanenbaum Research Institute (LTRI) in Mount Sinai Hospital for providing reagents and piggybac vectors; Michaal Woodside, Paul Paroutis and Kimberly Lau from Sickkids Imaging Facility for help in image acquisition and analysis; Andreas Helbig from the Lunenfeld-Tanenbaum Research Institute (LTRI) in Mount Sinai Hospital and Monika Tucholska from the Network Biology Collaborative Centre (NBCC) located at the Lunenfeld-Tanenbaum Research Institute (LTRI) in Mount Sinai Hospital for help with mass spectrometry analysis; Sheyun Zhao and Queenie

Cheung from Sickkids Flow Cytometry facility for assistance in flow cytometry analysis and cell sorting; The authors wish to acknowledge the contribution of the Model Production Core staff led by Marina Gertsenstein at The Centre for Phenogenomics for technical support; Jorge Cabezas from Toronto Centre of Phenogenomics for help in mouse colony management. We thank Dr. Eszter Posfai for critical reading and comments.

## Additional information

### Funding

| Funder | Grant reference number | Author |
| --- | --- | --- |
| Canadian Institutes of Health Research | FDN-143334 | Bin Gu<br>Katie Cockburn<br>Janet Rossant |
| Ontario Institute for Regenerative Medicine | | Bin Gu |
| Cancer Research Society | | Jean-Philippe Lambert |
| Canadian Institutes of Health Research | FDN143301 | Jean-Philippe Lambert<br>Anne-Claude Gingras |

The funders had no role in study design, data collection and interpretation, or the decision to submit the work for publication.

### Author contributions

BG, Conceptualization, Resources, Data curation, Formal analysis, Validation, Investigation, Visualization, Methodology, Writing—original draft, Writing—review and editing; J-PL, Resources, Data curation, Software, Formal analysis, Investigation, Visualization, Methodology, Writing—review and editing; KC, Investigation, Writing—review and editing; A-CG, Resources, Data curation, Funding acquisition, Writing—review and editing; JR, Conceptualization, Resources, Supervision, Funding acquisition, Writing—original draft, Writing—review and editing

### Author ORCIDs

Bin Gu, http://orcid.org/0000-0002-5594-2463
Janet Rossant, http://orcid.org/0000-0002-3731-5466

### Ethics

Animal experimentation: All animal work was carried out following Canadian Council on Animal Care Guidelines for Use of Animals in Research and Laboratory Animal Care under protocols approved by The Centre for Phenogenomics Animal Care Committee (protocol number: 20-0026H).

## Additional files

### Supplementary files

• Supplementary file 1. Reagents and resources used in this study

• Supplementary file 2. Oligos and primers sequences

### Major datasets

The following dataset was generated:

| Author(s) | Year | Dataset title | Dataset URL | Database, license, and accessibility information |
| --- | --- | --- | --- | --- |
| Gu B, Lambert JP, Cockburn K, Gingras AC, Rossant J | 2017 | AIRE BioID dataset | http://proteomecentral.proteomexchange.org/cgi/GetDataset?ID= | Publicly available at ProteomeXchange (accession no. |

PXD005529                    PXD005529)

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
