## [Decision Letter]

Thank you for submitting your article "AIRE is a critical spindle-associated protein in embryonic stem cells" for consideration by *eLife*. Your article has been reviewed by two peer reviewers, and the evaluation has been overseen by a Reviewing Editor and Fiona Watt as the Senior Editor. The following individual involved in review of your submission has agreed to reveal his identity: Peter W Andrews (Reviewer #2).

The reviewers have discussed the reviews with one another and the Reviewing Editor has drafted this decision to help you prepare a revised submission.

Summary:

The study provides evidence for a novel role for AIRE in organisation of the mitotic spindle in early development and in pluripotent stem cells. The findings are relevant to the maintenance of the genetic stability of pluripotent stem cells in vitro and to regulation of early embryo development.

Essential revisions:

Both reviewers express considerable enthusiasm for the work. Reviewer 1 raises several specific points that should be addressed by revision of the text.

"As a minor point, in the Abstract and Introduction, it might be helpful for some readers if it is explicitly made clear that the work is with mouse and not human ES cells. Also as the techniques of BioID and Duolink are not widely used, it would help uninitiated readers if the text included a very brief description of what they are."

Reviewer 2 highlights some areas where uncertainty regarding mechanisms remain. These should be addressed in the Discussion.

"the exact function of AIRE in the spindle complex or the mechanism by which AIRE stabilizes centrioles has not been entirely revealed. Along those lines, how does depletion of AIRE causes destabilization of the spindle machinery? Can the authors speculate what the reason is for the increased number of centrioles in the absence of AIRE? Are the centrioles not properly divided? Is there some kind of compensating mechanism?"

"As the authors noted, and as their immunofluorescence data shows, AIRE is present in cellular foci in addition to the spindle pole even during mitosis. This raises the issue that the interactome includes interacting partners which are disconnected from the spindle pole, and that phenotypes in the mutant cells cannot be unequivocally distinguished between the spindle phenotype and other cellular phonotypes."

Finally reviewer 2 notes that Figure 6 does not convincingly support claims of specific AIRE expression in the ICM. The figure should be replaced or the claims modified, and the relationship between expression in the embryo and TE or ES stem cell lines clarified.

"The authors note that AIRE staining in embryos was observed in both TE cells and in ICM cells (subsection “Aire is critical for spindle integrity in preimplantation embryos”). However, in the Introduction the authors state: ‘Among embryo-derived stem cell lines, Aire was specific to pluripotent ES cells and not found in extraembryonic trophoblast stem cells or extraembryonic endoderm cells (our unpublished data)’. In fact, looking at the presented image, the control AIRE staining in Figure 6 looks very dim, and the only cells which seem to be somewhat slightly AIRE positive are the TE cells. The ICM seems not to express AIRE at all. This should be clarified."

---

## [Author Response]

*Essential revisions:*

*Both reviewers express considerable enthusiasm for the work. Reviewer 1 raises several specific points that should be addressed by revision of the text.*

*"As a minor point, in the Abstract and Introduction, it might be helpful for some readers if it is explicitly made clear that the work is with mouse and not human ES cells. Also as the techniques of BioID and Duolink are not widely used, it would help uninitiated readers if the text included a very brief description of what they are."*

We thank reviewer 1 for pointing out that we haven’t made explicitly clear that the work is with mouse and not human ES cells. We have edited the Abstract and Introduction section to clearly state that the works presented in this study were performed on mouse ES cells and mouse embryos (Abstract; Introduction: second and last paragraphs).

We thank reviewer 1 for suggesting that it would be helpful to include a brief description of BioID and Duolink technology. We edited the text in Introduction to present a more clear description of BioID (Introduction: third paragraph). We also included a brief description of Duolink technology in the Materials and methods subsection “Proximity Ligation Amplification(PLA) Assay”.

Reviewer 2 highlights some areas where uncertainty regarding mechanisms remain. These should be addressed in the Discussion.

*"the exact function of AIRE in the spindle complex or the mechanism by which AIRE stabilizes centrioles has not been entirely revealed. Along those lines, how does depletion of AIRE causes destabilization of the spindle machinery? Can the authors speculate what the reason is for the increased number of centrioles in the absence of AIRE? Are the centrioles not properly divided? Is there some kind of compensating mechanism? "*

We thank reviewer 2 Dr. Andrews for pointing out that it would be valuable to speculate the mechanism by which Aire depletion or dominant negative overexpression caused the mitotic phenotypes reported here. The speculated mechanisms are addressed in Discussion section as:

‘The mechanism by which Aire sustained the centrosome number stability and structure integrity in mES cells is not clear at this point. […] Disruption of this interaction could also contribute to the centrosome phenotypes of Aire depletion and AIRE^ΔLESLL^ overexpression in mES cells.’

And the speculation that Deaf1 gene could be a redundant gene with Aire in mES cells were addressed in Discussion section as:

“Another possible explanation for the observation of more severe mitotic phenotype in AIRE^ΔLESLL^ overexpressing cells than Aire depleted cells is the existence of redundant pathways. […] It would be interesting to study the effect of depletion of Deaf1 or combined depletion of Aire and Deaf1 to understand the function of this group of genes in mES cells.”

*"As the authors noted, and as their immunofluorescence data shows, AIRE is present in cellular foci in addition to the spindle pole even during mitosis. This raises the issue that the interactome includes interacting partners which are disconnected from the spindle pole, and that phenotypes in the mutant cells cannot be unequivocally distinguished between the spindle phenotype and other cellular phonotypes. "*

We thank reviewer 2 Dr. Andrews for pointing out and agree that at this point we could not unequivocally distinguish between a direct effect on spindle assembly and indirect effects from defects of other processes that Aire participates in in mES cells. We addressed this in Discussion section as:

“However, since AIRE also interacted with proteins in mES cells that function in processes such as transcription regulation and RNA processing, it is still possible that defects in those processes upon Aire depletion or AIRE^ΔLESLL^ overexpression indirectly contributed to the mitotic phenotype. Further study is required to separate out the different functions of Aire in mES cells and embryos.”

*Finally reviewer 2 notes that Figure 6 does not convincingly support claims of specific AIRE expression in the ICM. The figure should be replaced or the claims modified, and the relationship between expression in the embryo and TE or ES stem cell lines clarified.*

*"The authors note that AIRE staining in embryos was observed in both TE cells and in ICM cells (subsection “Aire is critical for spindle integrity in preimplantation embryos”). However, in the Introduction the authors state: ‘Among embryo-derived stem cell lines, Aire was specific to pluripotent ES cells and not found in extraembryonic trophoblast stem cells or extraembryonic endoderm cells (our unpublished data)’. In fact, looking at the presented image, the control AIRE staining in Figure 6 looks very dim, and the only cells which seem to be somewhat slightly AIRE positive are the TE cells. The ICM seems not to express AIRE at all. This should be clarified."*

We thank reviewer 2 Dr. Andrews for pointing out the confusion about the contradiction between the claim of Epiblast specific expression in embryos and the pan-blastocyst expression showed Figure 6 and we totally agree that we were not clear in the statements. In fact, we didn’t try to claim Epiblast specific expression of Aire in mouse early embryos. In Introduction we did report that we observed specific expression of Aire in mouse ES cells relative to TS and Xen cells. However, TS cells to a larger extend represents post-implantation extraembryonic-ectoderm(EXE) progenitor cells (now stated in Introduction section, second paragraph) and the molecular characteristics are rather different from preimplantation TE. For example, TS cells highly express Sox2 while TE cells don’t and TS cells highly express Elf5 while TE cells only express it at very low level. Therefore, we actually only claimed that Aire was differentially expressed between the three types of in vitro stem cell lines but didn’t make any claim about the expression pattern in mouse preimplantation embryos/blastocysts. To further clarify it, we stated in Introduction section that ‘The presence of Aire mRNA have been detected in mouse oocytes and all preimplantational stages and early postimplantational stages (up till E6.5) embryos but the expression levels and patterns at protein level are unknown.”

Therefore to our knowledge our immunostaining data in Figure 6 is the first strictly controlled data (with a m-z- staining control) that revealed the expression pattern of AIRE in mouse blastocysts at protein level. We agree that the protein level of AIRE in mouse blastocysts is relatively low, however, with exactly same imaging acquisition and displaying parameters, it showed obvious positive signals in both ICM and TE relative to the m-z- embryos. We apologize for the dim looking pictures on original Figure 6, probably caused by the image transformation to PDF. We have replaced the picture with a more clear version.